# Histopathologic image–based deep learning classifier for predicting platinum-based treatment responses in high-grade serous ovarian cancer

Byungsoo Ahn [1,13], Damin Moon [2,13], Hyun-Soo Kim [3,13], Chung Lee[1], Nam Hoon Cho[1], Heung-Kook Choi[2], Dongmin Kim [2], Jung-Yun Lee [4], Eun Ji Nam[4], Dongju Won[5], Hee Jung An[6], Sun Young Kwon[7], Su-Jin Shin[8], Hye Ra Jung[7], Dohee Kwon[1], Heejung Park[1], Milim Kim[1], Yoon Jin Cha[8,9], Hyunjin Park[8], Yangkyu Lee[8], Songmi Noh[10], Yong-Moon Lee[11], Sung-Eun Choi[6], Ji Min Kim[12], Sun Hee Sung[12] & Eunhyang Park [1] ✉

Platinum-based chemotherapy is the cornerstone treatment for female high-grade serous ovarian carcinoma (HGSOC), but choosing an appropriate treatment for patients hinges on their responsiveness to it. Currently, no available biomarkers can promptly predict responses to platinum-based treatment. Therefore, we developed the Pathologic Risk Classifier for HGSOC (PathoRiCH), a histopathologic image–based classifier. PathoRiCH was trained on an in-house cohort ($n = 394$) and validated on two independent external cohorts ($n = 284$ and $n = 136$). The PathoRiCH-predicted favorable and poor response groups show significantly different platinum-free intervals in all three cohorts. Combining PathoRiCH with molecular biomarkers provides an even more powerful tool for the risk stratification of patients. The decisions of PathoRiCH are explained through visualization and a transcriptomic analysis, which bolster the reliability of our model's decisions. PathoRiCH exhibits better predictive performance than current molecular biomarkers. PathoRiCH will provide a solid foundation for developing an innovative tool to transform the current diagnostic pipeline for HGSOC.

Epithelial ovarian cancer is the most common gynecological malignancy and the eighth leading cause of cancer-related deaths among females worldwide[1]. Most epithelial ovarian cancers are high-grade serous ovarian carcinoma (HGSOC), characterized by advanced stages (III and IV) at initial diagnosis, rapid progression with widespread dissemination, and poor prognosis. For patients with advanced HGSOC, the 5-year survival rate is approximately 25%[2]. Platinum-based chemotherapy following debulking surgery is the standard treatment for HGSOC. However, clinical responses to platinum therapy vary[3–5]: only 20% of advanced-stage patients show a favorable treatment response and long-term survival, whereas the remaining 80% relapse within two years and are left with limited treatment options[4,5].

Poly ADP-ribose polymerase (PARP) inhibitors, which exploit DNA repair vulnerabilities, have rapidly become game changers in ovarian cancer treatment[6–8]. The best predictors of sensitivity to PARP inhibitors are the platinum-treatment response, *BRCA1/2* (*BRCA*) mutation status, and homologous recombination deficiency (HRD) status. These predictors have limitations, however. To determine whether an

individual is platinum-sensitive or -resistant, patients must undergo several cycles of chemotherapy and experience related adverse events. Although various genomic, transcriptomic, and proteomic biomarkers have been proposed to predict the outcomes of platinum-based chemotherapy, none has yet been introduced into standard clinical practice[6,8–10]. In addition, genomic assays for *BRCA* mutations and HRD status are expensive, entail a long turnaround time, and require tumor DNA/RNA samples for analysis, making them challenging to implement in every patient with HGSOC, especially in low-resource settings[11,12].

To initiate adjuvant chemotherapy for HGSOC, a pathological diagnosis based on hematoxylin and eosin (H&E)-stained whole slide images (WSIs) is essential. Those histopathological images might contain vital information about the biological behaviors of tumors and could be critical for predicting chemo-responsiveness. HGSOC exhibits various histopathologic features, but so far, no one has identified pathologic factors that predict clinical outcomes. Recently, the combination of deep learning and digital pathologic imaging has made it possible to automate routine diagnostic tasks, such as cancer detection, grading, and subtyping[13,14]. In addition, it has provided a way to discover previously unrecognized prognostic morphological traits that can be used to predict treatment responses and outcomes or to infer the molecular characteristics of tumors[15–21]. Although several studies have tried to predict the clinical outcomes or molecular features of HGSOC from histopathological tumor images, most had small sample sizes, lacked external validation, or did not demonstrate the reliability of their models[22–26]. Therefore, the predictive power of histology in HGSOC needs to be further investigated.

In this work, we used various multiple instance learning (MIL) models that use only histopathological images to predict responses to platinum-based treatment in female HGSOC. In doing so, we were able to develop a robust MIL model that we call the Pathologic Risk Classifier for HGSOC (PathoRiCH). PathoRiCH was trained with HGSOC cohort and it exhibits significant predictive performance. In addition, combining PathoRiCH with current molecular biomarkers provides an even more powerful risk stratification method for patients with HGSOC. To demonstrate the reliability of our model, we visualized the model outputs and analyzed the molecular characteristics of the predicted groups. Application of this model will provide information that is clinically relevant for guiding patient-tailored therapy in HGSOC.

## Results
### Cohort characteristics
We analyzed 814 patients with HGSOC: 394 patients (WSI $n = 754$) treated at Yonsei Severance Hospital (SEV cohort), 284 patients (WSI $n = 516$) from the Cancer Genome Atlas Ovarian Cancer (TCGA-OV) database (TCGA cohort), and 136 patients (WSI $n = 136$) treated at Samsung Medical Center (SMC cohort). The clinicopathological characteristics of each cohort are summarized in Table 1. The patients were classified into four groups according to their platinum-free interval (PFI), the time between the last platinum-based chemotherapy cycle and the first recurrence: "platinum-resistant" (PFI ≤ 6 months), "partially platinum-sensitive" (PFI 6–12 months), "platinum-sensitive" (PFI 12–24 months), and "very platinum-sensitive" (PFI > 24 months)[27,28]. To predict the responses to platinum-based therapy, we used 12-month cut-off and classified "platinum-resistant" and "partially platinum-sensitive" patients as the poor response group and "platinum-sensitive" and "very platinum-sensitive" patients as the favorable response group. For *BRCA* mutation and HRD status prediction, patients with available *BRCA* mutation or HRD status results ($n = 767$ and $n = 284$, respectively) were evaluated.

### MIL models
We investigated six different MIL models using two image areas (all-tissue and cancer-segmented areas) and three magnifications (5×, 20×, and a combination of 5× and 20×). An overview of the proposed MIL model is presented in Fig. 1. For the cancer-segmented area, a cancer segmentation model pretrained with invasive breast ductal carcinoma automatically labeled the cancer areas (Supplementary Fig. 1). The cancer-segmented area demonstrated good overall concordance with the pathologist-annotated cancer area across both the internal (SEV) and external (TCGA) cohorts, achieving Dice coefficients of 0.781 and 0.836, respectively (Supplementary Table 1 and Supplementary Fig. 2). For the all-tissue and cancer-segmented area models, 17,742,605 and 3,822,597 patches were trained, respectively. The image magnification settings were based on the usual approach to pathological diagnoses: low-magnification (5×) for architectural-level evaluation, high-magnification (20×) for cytologic-level evaluation, and multiscale levels to integrate information from the 5× and 20× images.

Five-fold cross-validation of the SEV cohort was used for training and internal validation. In the internal (SEV) validation, the all-tissue area MIL generally showed better performance than the cancer-segmented area MIL (Table 2). Specifically, the all-tissue area 5×

## Table 1 | Patient characteristics of all cohorts in high-grade serous ovarian carcinoma

| | All cohorts (N = 814) | | |
|---|---|---|---|
| | **SEV** (N = 394) | **TCGA** (N = 284) | **SMC** (N = 136) |
| Number of WSIs | 754 | 516 | 136 |
| Age | 53.9 ± 10.9 | 59.8 ± 11.2 | 56.9 ± 8.7 |
| Stage | | | |
| I | 0 (0.0%) | 12 (4.2%) | 12 (8.8%) |
| II | 0 (0.0%) | 24 (8.5%) | 14 (10.3%) |
| III | 214 (54.3%) | 227 (79.9%) | 77 (56.6%) |
| IV | 180 (45.7%) | 20 (7.0%) | 33 (24.3%) |
| Not available | 0 (0.0%) | 1 (0.4%) | 0 (0.0%) |
| *BRCA* mutation status | | | |
| Mutant | 65 (16.5%) | 19 (6.7%) | 28 (20.6%) |
| Wildtype | 148 (37.6%) | 265 (93.3%) | 107 (78.7%) |
| Unknown | 181 (45.9%) | 0 (0.0%) | 1 (0.7%) |
| HRD status (Telli et al.)[33] | | | |
| Positive | 0 (0.0%) | 153 (55.4%) | 0 (0.0%) |
| Negative | 0 (0.0%) | 114 (41.3%) | 0 (0.0%) |
| Unknown | 0 (0.0%) | 9 (3.3%) | 0 (0.0%) |
| HRD status (Takaya et al.)[34] | | | |
| Positive | 0 (0.0%) | 140 (49.3%) | 0 (0.0%) |
| Negative | 0 (0.0%) | 139 (48.9%) | 0 (0.0%) |
| Unknown | 0 (0.0%) | 5 (1.8%) | 0 (0.0%) |
| HRD status (Perez-Villatoro et al.)[35] | | | |
| Positive | 0 (0.0%) | 68 (23.9%) | 0 (0.0%) |
| Negative | 0 (0.0%) | 15 (5.3%) | 0 (0.0%) |
| Not evaluated | 0 (0.0%) | 29 (10.2%) | 0 (0.0%) |
| Undefined | 0 (0.0%) | 172 (60.6%) | 0 (0.0%) |
| Platinum response groups | | | |
| Platinum-resistant (PFI < 6 mo) | 73 (18.5%) | 12 (4.2%) | 6 (4.4%) |
| Partially platinum-sensitive (PFI 6–12 mo) | 59 (15.0%) | 47 (16.5%) | 32 (23.5%) |
| Platinum-sensitive (PFI 12–24 mo) | 85 (21.6%) | 128 (45.1%) | 2 (1.5%) |
| Very platinum-sensitive (PFI > 24 mo) | 177 (44.9%) | 97 (34.2%) | 96 (70.6%) |

*HRD* homologous recombination deficiency, *mo* months, *PFI* platinum-free interval, *WSIs* whole slide images.

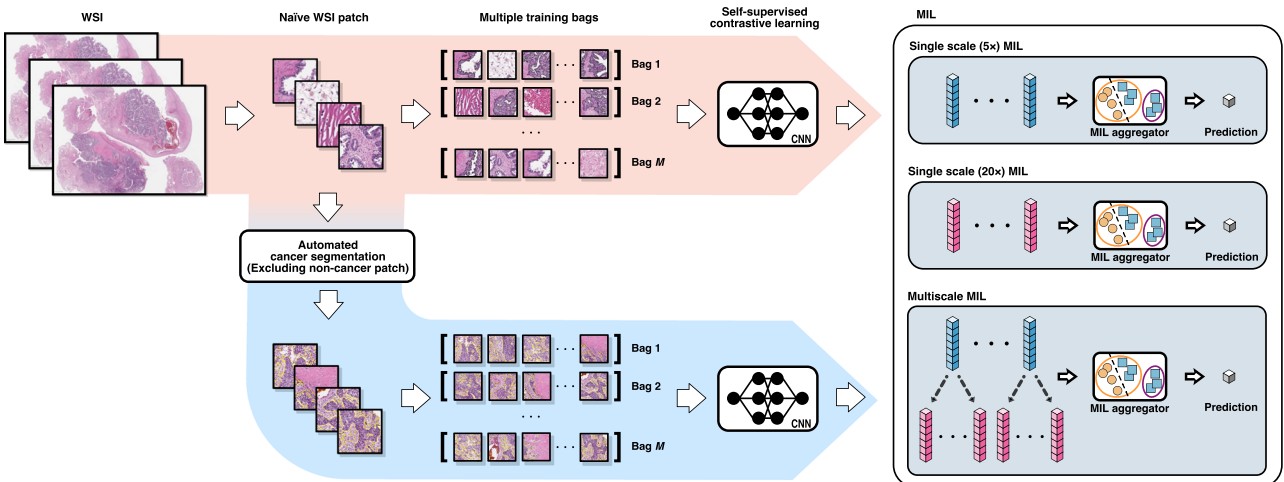

**Fig. 1 | Overview of our multiple instance learning models.** Patches of varying magnifications (5× and 20×) were extracted from the whole-slide images (WSIs). The patches were then processed using automated cancer segmentation to exclude patches without cancer cells and fed into a contrastive self-supervised learning algorithm (blue arrow path). Alternatively, all patches, including those without cancer cells, could be fed directly into the self-supervised learning algorithm to include all tissues in the WSIs (red arrow path). Separate multiple instance learning (MIL) methods were used for two single scales and one multiscale magnification setting (5×, 20×, and both) for each image area. Therefore, six different MIL models were generated. For the multiscale MILs, feature pyramids were formed by concatenating the embeddings of different scales of WSIs to train the MIL aggregator.

**Table 2 | Performance of multiple instance learning models in internal (SEV) and external (TCGA and SMC) validation cohorts in predicting platinum-based treatment response groups**

| | | All-tissue area MIL | | | Cancer-segmented area MIL | | |
|---|---|---|---|---|---|---|---|
| | | 5× | 20× | Multiscale | 5× | 20× | Multiscale |
| Internal validation (SEV cohort) | AUC-ROC[a] | 0.627 ± 0.047 | 0.610 ± 0.04 | 0.623 ± 0.016 | 0.604 ± 0.05 | 0.596 ± 0.072 | 0.614 ± 0.046 |
| | Precision | 0.495 | 0.605 | 0.565 | 0.521 | 0.465 | 0.507 |
| | Recall | 0.663 | 0.411 | 0.545 | 0.468 | 0.675 | 0.525 |
| | F1 score | 0.559 | 0.462 | 0.517 | 0.470 | 0.522 | 0.507 |
| | K-M $p$ value[b] | 0.000 | 0.000 | 0.000 | 0.000 | 0.000 | 0.000 |
| External validation (TCGA cohort) | AUC-ROC | 0.492 | 0.594 | 0.575 | 0.532 | 0.602 | 0.573 |
| | Precision | 0.187 | 0.253 | 0.232 | 0.519 | 0.406 | 0.407 |
| | Recall | 0.879 | 0.484 | 0.429 | 0.250 | 0.528 | 0.481 |
| | F1 score | 0.309 | 0.332 | 0.301 | 0.338 | 0.459 | 0.441 |
| | K-M $p$ value[b] | 0.108 | 0.004 | 0.000 | 0.000 | 0.032 | 0.036 |
| External validation (SMC cohort) | AUC-ROC | - | - | - | - | 0.593 | - |
| | Precision | - | - | - | - | 0.351 | - |
| | Recall | - | - | - | - | 0.711 | - |
| | F1 score | - | - | - | - | 0.470 | - |
| | K-M $p$ value[b] | - | - | - | - | 0.030 | - |

*AUC-ROC* area under the receiver operating characteristic curve, *K-M* Kaplan–Meier analysis (two-sided), *MIL* multiple instance learning.
[a]From 5-fold cross validation.
[b]Based on platinum-free interval.

magnification model showed the best performance, achieving an average area under the receiver operating characteristic curve (AUC-ROC) value of 0.627. However, in the external (TCGA) validation, the cancer-segmented area 20× magnification model showed the best performance, with AUC-ROC values of 0.602, and the performance of the 5× models showed the largest decrease among the all-tissue and cancer-segmented area MILs. The multiscale models exhibited intermediate performance between the 5× and 20× models in both the all-tissue and cancer-segmented area MILs. To confirm that result, we performed ensemble techniques using both 5× and 20× images of the cancer-segmented areas (Supplementary Table 2). Compared with the soft and hard voting ensemble models, the cancer-segmented area 20× magnification model showed superior performance.

## PathoRiCH + BRCA + HRD shows the best PFI prediction ability

To achieve a balanced good performance across the internal and external validations, we opted for the cancer-segmented area 20× magnification MIL and termed it PathoRiCH. In the Kaplan–Meier analysis, the favorable and poor groups determined by PathoRiCH in the internal (SEV) and external (TCGA) validation cohorts showed significantly differentiated PFIs ($p < 0.001$ and $p = 0.032$, respectively) (Fig. 2a, b). PathoRiCH also showed significantly different distributions of the four PFI groups (PFI ≤ 6 months, 6–12 months, 12–24 months, and >24 months) in the two cohorts: $p = 0.036$ and $p < 0.001$, respectively (Supplementary Fig. 3). When the PathoRiCH results were combined with the *BRCA* and HRD status (PathoRiCH+*BRCA* + HRD) in the TCGA cohort, the patients could be further stratified into four

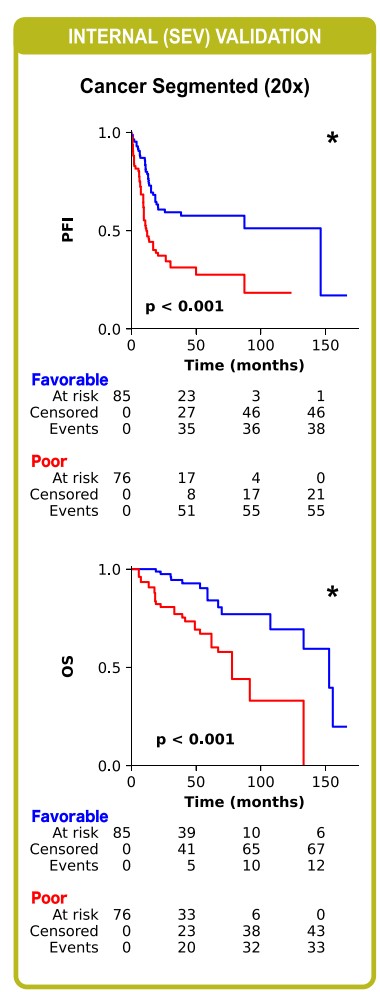

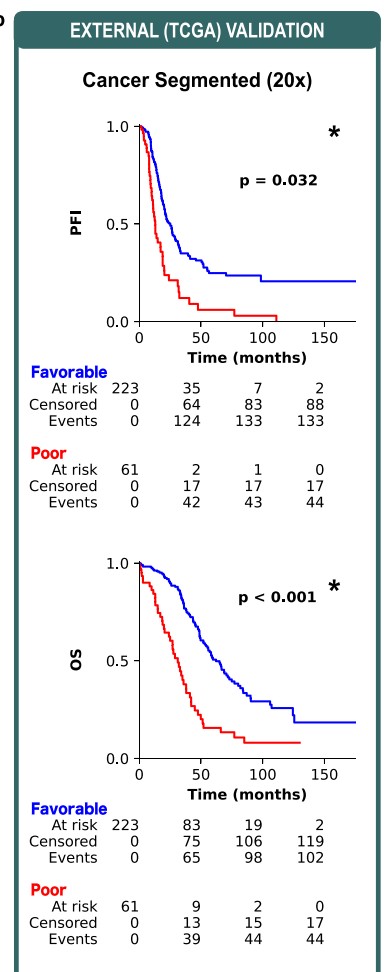

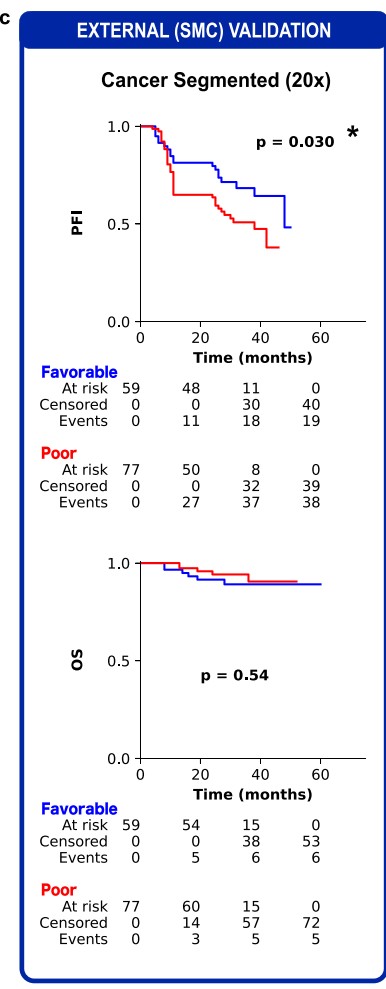

**Fig. 2 | Kaplan–Meier survival analysis of the cancer-segmented area 20× magnification multiple instance learning model (PathoRiCH).** Two-sided Kaplan–Meier survival analysis was used. **a** In the internal validation, the PathoRiCH-predicted favorable and poor groups exhibited significant differences in the platinum-free interval (PFI) and overall survival (OS) ($p = 4.17E\text{-}05$ and $p = 8.73E\text{-}05$, respectively). **b** Analysis of the TCGA external validation cohort revealed significant patient stratification for PFI ($p = 0.032$) and OS ($p = 1.06E\text{-}09$). **c** The SMC external validation cohort also showed significant patient stratification for PFI ($p = 0.030$), but it did not reach statistical significance for OS ($p = 0.54$).

subgroups, favorable–*BRCA*/HRD-positive (47.8%), favorable–*BRCA*/HRD-negative (31.7%), poor–*BRCA*/HRD-positive (10.4%), and poor–*BRCA*/HRD-negative (10.1%), enabling more precise risk stratification than using only molecular biomarkers. The favorable–*BRCA*/HRD-positive group showed the best PFI, the favorable–*BRCA*/HRD-negative group showed moderate PFI, and the poor–*BRCA*/HRD-positive and poor–*BRCA*/HRD-negative groups showed the worst PFI ($p < 0.001$) (Fig. 3a). Furthermore, PathoRiCH+*BRCA* + HRD–defined subgroups showed significantly different distributions of the four PFI groups ($p = 0.001$) (Fig. 3b). For comparison, the Kaplan–Meier analyses for the ground truth favorable and poor response groups in the internal (SEV) and external (TCGA) validation cohorts are shown in Supplementary Fig. 4.

We next assessed PathoRiCH's performance in an additional independent external (SMC) validation cohort, and PathoRiCH yielded an AUC-ROC value of 0.593 (Table 2). In SMC cohort, the PathoRiCH-predicted favorable and poor response groups also demonstrated a statistically significant difference in PFIs ($p = 0.030$) (Fig. 2c) and significantly different distributions of the four PFI groups ($p < 0.001$) (Supplementary Fig. 3). Because the SMC dataset does not provide HRD results, the PathoRiCH+*BRCA* + HRD combination was not available. For comparison, the Kaplan–Meier analyses for the ground truth favorable and poor groups is shown in Supplementary Fig. 4.

## PathoRiCH was identified as an independent prognostic factor

In the external (TCGA) validation cohort, the PathoRiCH-predicted groups exhibited no significant associations with age, *BRCA* mutation status, and HRD status, but FIGO stage was significantly associated ($p < 0.001$) (Supplementary Table 3). In univariate Cox regression analyses, PathoRiCH exhibited the strongest association with PFI ($p < 0.0001$), followed by FIGO stage ($p = 0.001$), *BRCA* status ($p = 0.026$), HRD status ($p = 0.010$), and *BRCA* + HRD status ($p = 0.011$) (Supplementary Table 4). Age, however, did not show a significant association with PFI. In a multivariate Cox regression analysis, PathoRiCH was identified as the strongest independent prognostic factor ($p < 0.0001$), with a hazard ratio of 1.947 (95% confidence interval = 1.350–2.808, $p < 0.001$), and FIGO stage and *BRCA* status were also identified as independent prognostic factors ($p = 0.005$ and $p = 0.32$, respectively) (Fig. 4a). The Kaplan–Meier plots and distributions of the four PFI groups according to *BRCA* mutation and HRD status are shown in Supplementary Fig. 5a.

In the external SMC validation cohort, the PathoRiCH-predicted groups displayed no significant associations with any of the clinical or molecular variables (Supplementary Table 3). In univariate Cox regression analyses, similar to the TCGA cohort, FIGO stage ($p = 0.003$), *BRCA* status ($p = 0.026$), and PathoRiCH ($p = 0.027$) displayed statistical significance (Supplementary Table 4). Of them, the

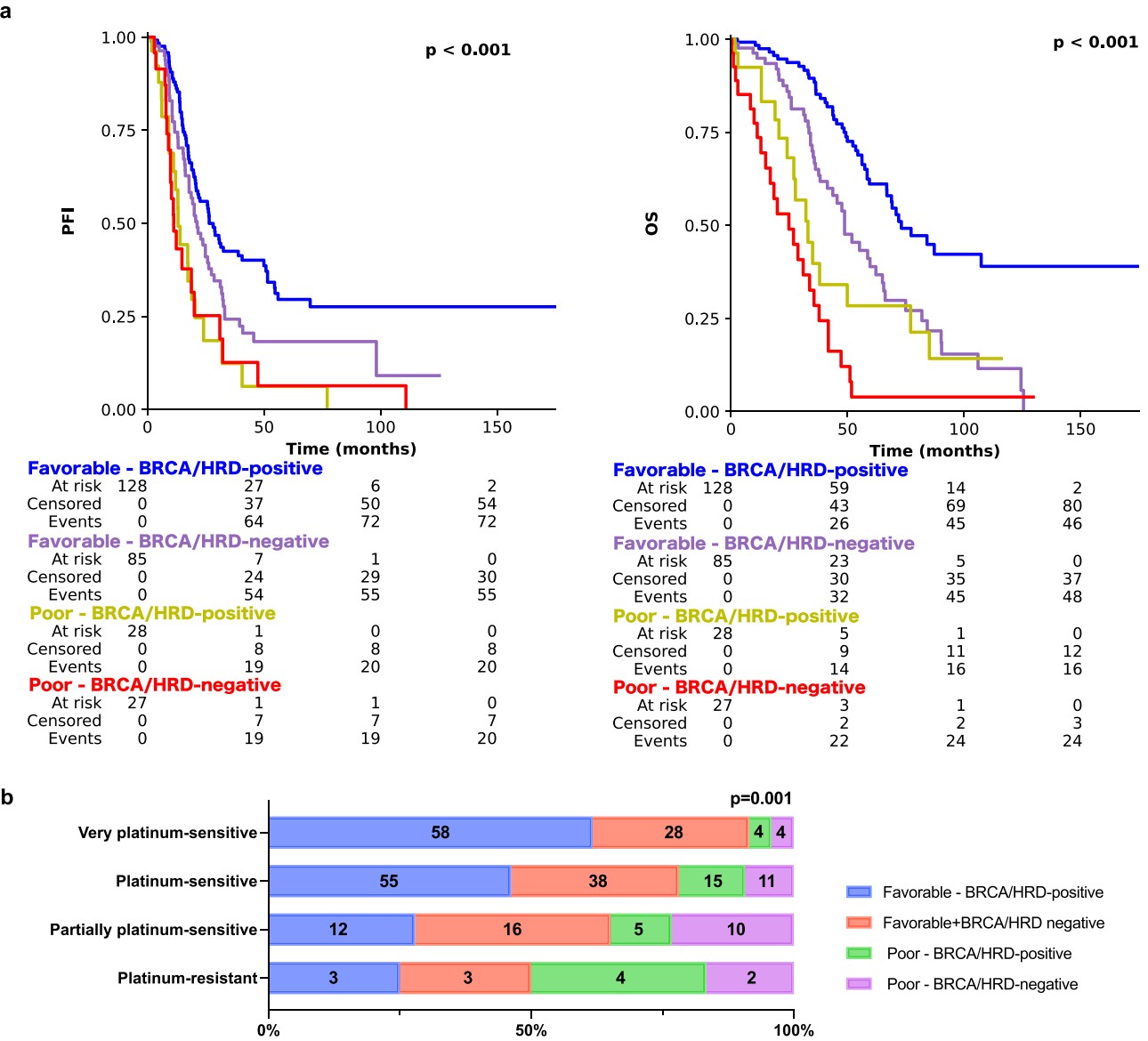

**Fig. 3 | Kaplan–Meier survival analyses and distribution of the true platinum-free interval groups of PathoRiCH + BRCA + HRD in the TCGA external validation cohort.** **a** Kaplan–Meier survival plots of patients categorized by combined PathoRiCH, *BRCA*, and HRD results. The combined PathoRiCH, *BRCA*, and HRD significantly differentiated response groups in the platinum-free interval (PFI) and overall survival (OS) ($p = 1.07E-05$ and $p = 3.30E-16$, respectively). The favorable–*BRCA*/HRD-positive group displayed the most favorable PFI, and the poor–*BRCA*/HRD-positive and poor–*BRCA*/HRD-negative groups showed the worst PFI. Two-sided Kaplan–Meier survival analysis was used. **b** Distribution of the four PFI groups (platinum resistant (PFI ≤ 6 months), partially platinum resistant (6–12 months), platinum sensitive (12–24 months), and very platinum sensitive (>24 months)) by combined PathoRiCH, *BRCA*, and HRD. The colored bars indicate the percentage of predictions for each outcome group (blue for favorable and red for poor), with numerical values within each bars showing the case count for each category. The combined PathoRiCH+*BRCA* + HRD showed significantly different distributions for the four PFI groups ($p = 0.001$).

FIGO stage ($p = 0.003$) and PathoRiCH predictions ($p = 0.038$) were identified as independent prognostic factors (Fig. 4b). The Kaplan–Meier plots and distributions of the four PFI groups according to *BRCA* mutation are shown in Supplementary Fig. 5b.

**Visualization analysis**

Because PathoRiCH makes decisions based on the predicted probabilities of being in the favorable or poor group, we generated three attention maps: for the favorable group prediction, poor group prediction, and combined prediction (Fig. 5). Based on the hypothesis that patients with extreme treatment responses might harbor more predictive histologic features than those with midrange responses, we extracted high-scoring patches for the 40 cases (WSI $n = 76$) with highly favorable and poor response in the SEV cohort (patch $n = 3500$, respectively). By applying the Gaussian mixture model (GMM)

clustering algorithm and pathologists' evaluations, we classified the high-scoring patches for the favorable and poor groups into four histologically distinct clusters, respectively. In the favorable group, unique clusters showing "intratumoral lymphocytic infiltration" and "small monotonous nuclei with high cellularity" were identified (Fig. 6a). Conversely, "hyperchromatic nuclei with poor cohesiveness and spindling features" and "cytoplasmic vacuoles and microcystic change" were distinct histologic features in the poor group (Fig. 6b). The remaining clusters showed overlapping histologic features in both groups.

When the high-scoring patches for both groups were mixed and clustered together, six histologically distinct clusters with different proportions in the favorable and poor groups were identified (Supplementary Table 5). Of them, the clusters characterized as "intratumoral lymphocytic infiltration" were

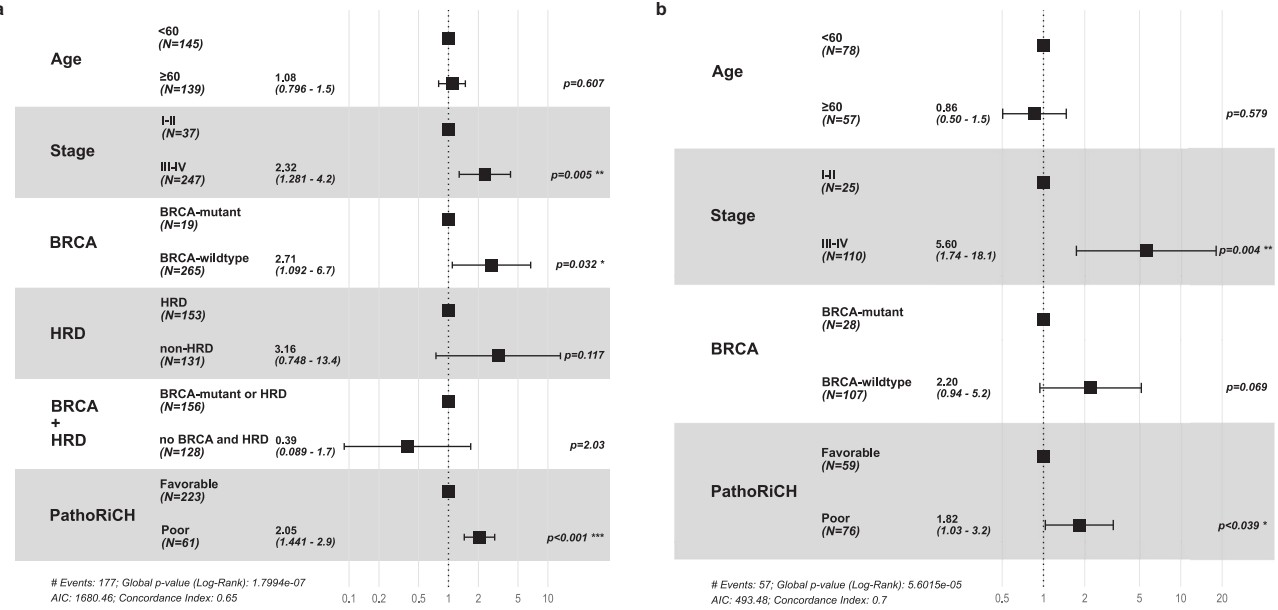

**Fig. 4 | Multivariate Cox regression analyses in the TCGA and SMC external validation cohorts.** The multivariate Cox regression analysis was conducted using six variables: age, FIGO stage, *BRCA* mutation status, HRD status, combined *BRCA* and HRD status, and PathoRiCH prediction. The data are presented with error bar representing 95% confidence interval. **a** In the TCGA cohort, PathoRiCH stood out as the most powerful independent prognostic factor (*p* = 6.57E-05), followed by FIGO stage (*p* = 0.005) and *BRCA* status (*p* = 0.32). **b** In the SMC cohort, FIGO stage (*p* = 0.004) and PathoRiCH (*p* = 0.39) stood out as significant independent prognostic factors.

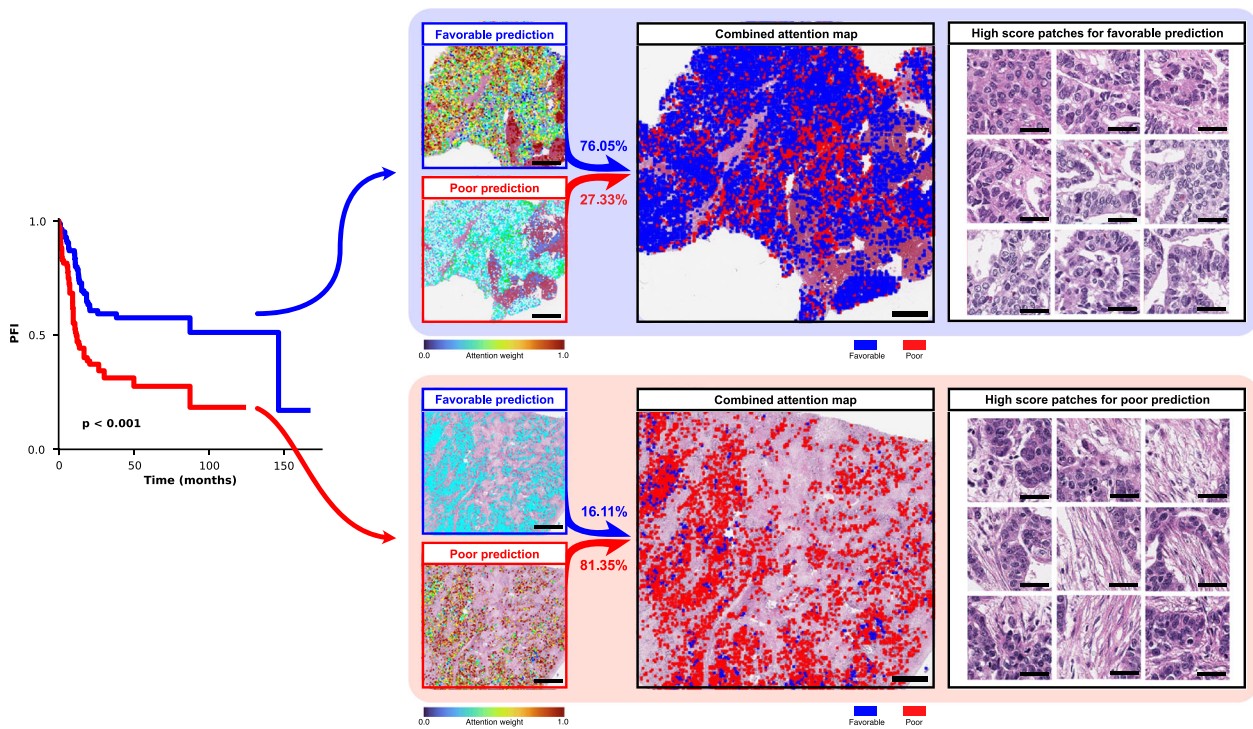

**Fig. 5 | Attention map analysis of the PathoRiCH-predicted favorable and poor groups.** The left side presents a two-sided Kaplan–Meier survival analysis according to PathoRiCH predictions. For these predictions, separate attention maps of favorable and poor predictions were created and then combined to generate a combined prediction attention map for each patient (scale bar = 2 mm). The figure shows two representative cases of patients from the favorable and poor groups, with the corresponding attention maps and high-score patches displayed side-by-side (scale bar = 50 μm).

identified as favorable group–dominant (favorable patches >80%), and the distinct histologic features in the poor group, "hyperchromatic nuclei with poor cohesiveness and spindling features" and "cytoplasmic vacuoles and microcystic change," were identified as poor group–dominant (poor patches >80%) (Fig. 6c).

The 100 highest scoring patches from the PathoRiCH-predicted favorable and poor groups are shown in Supplementary Fig. 6a.

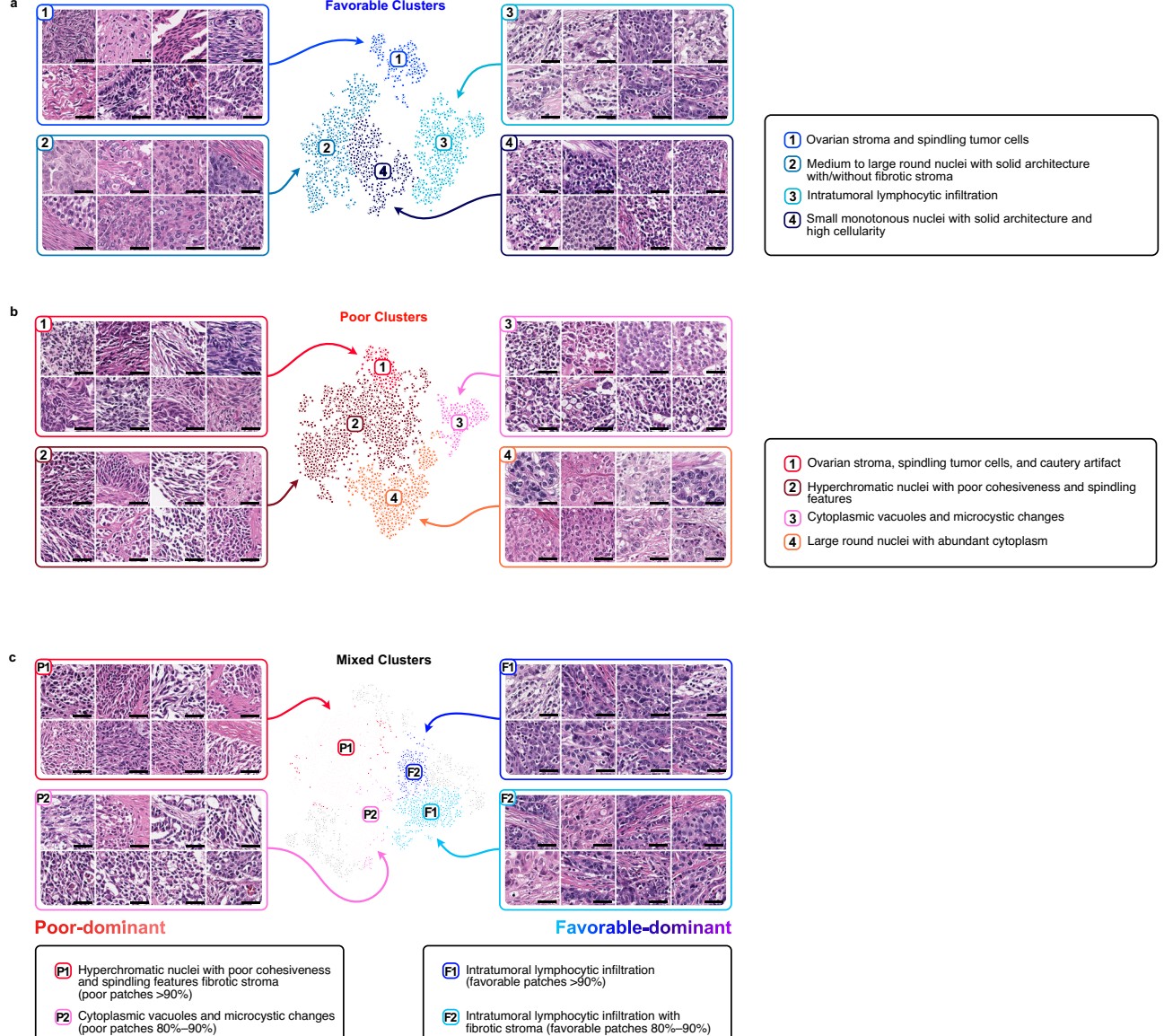

**Fig. 6 | Cluster analysis of high-score patches from the PathoRiCH-predicted favorable and poor groups.** (Scale bar = 50 μm for all patch images) (**a**, **b**) Clusters were initially created using Gaussian mixture models (GMMs), with high-score patches serving as inputs for each group. The resulting clusters were then evaluated by pathologists, who further combined clusters with similar histological features.

The final grouping comprised four favorable and four poor histologically distinct clusters. **c** The combination of high-score patches from the favorable and poor predicted groups was clustered based on their histological similarities using GMM. Seven clusters were created, and two favorable group–dominant clusters and two poor group–dominant clusters were identified.

Clusters showing "intratumoral lymphocytic infiltration" were included in the favorable group, and clusters showing "hyperchromatic nuclei with poor cohesiveness and spindling features" were frequently identified in the poor group. On the other hand, in the all-tissue area 20× models, the 100 highest scoring patches consisted mostly of non-tumor patches, such as adipose tissue and edge areas of tissue, which might have disturbed the model decision (Supplementary Fig. 6b).

**Transcriptome analysis**
For the 208 cases of the TCGA cohort with available RNAseq results, the PathoRiCH-predicted favorable and poor groups could be further divided into true-predicted (correct) and false-predicted (incorrect), as shown in Supplementary Table 6. First, we compared RNA expression patterns of the "true favorable-predicted" ($n = 134$) and "false favorable-predicted" ($n = 25$) groups. In differentially expressed gene (DEG) analysis (absolute log2 fold change >1 and $p < 0.01$), 13 up-regulated

genes and 25 down-regulated genes for the "true favorable-predicted" group were identified compared to "false favorable-predicted" group (Supplementary Fig. 7a and Supplementary Data 1). Notably, a subset of up-regulated genes, such as *PRSS16, KLKB1*, and *ACOD1*, were associated with immune response and immunometabolism[29,30]. A gene ontology (GO) analysis further identified enrichment of immune response–related genes in the "true favorable-predicted" group, while ribosomal- and mitochondrial-associated genes were prevalent in the "false favorable-predicted" group (Fig. 7a).

Next, when comparing of the "true poor-predicted" ($n = 19$) and "false poor-predicted" ($n = 30$) groups, 17 up-regulated genes and 26 down-regulated genes were identified for the "true poor-predicted" group (Supplementary Fig. 7b and Supplementary Data 2). Of these, stromal tissue related genes, such as *MYO16, ANKRD2, LRRC14B*, and *MYO7B*, were up-regulated in the "true poor-predicted" group[31,32]. In GO analysis, the "true poor-predicted" group also showed enriched in

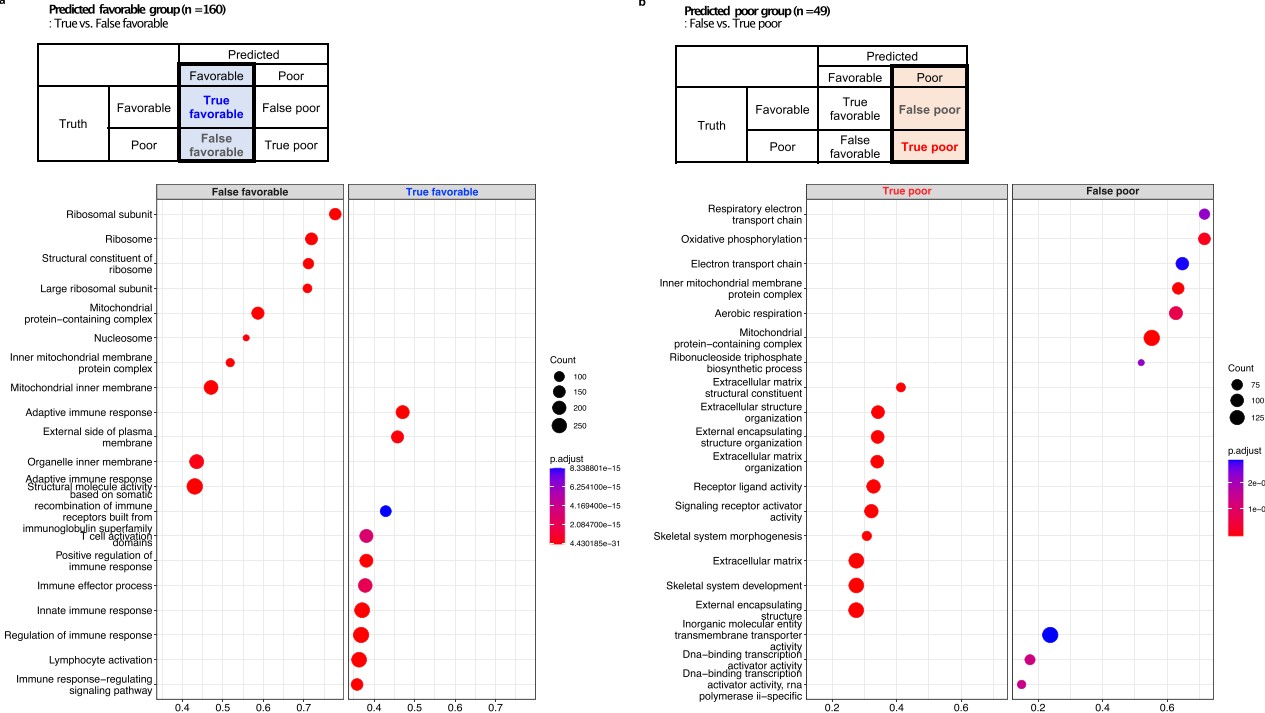

**Fig. 7 | Differential gene ontology (GO) profiles comparing True versus False classifications within the PathoRiCH-predicted favorable and poor outcome groups. a** Within the favorable outcome predictions, the true favorable-predicted ($n = 134$) predominantly featured genes involved in immune response, and the false favorable-predicted ($n = 25$) category was enriched for ribosomal- and mitochondrial-associated genes. **b** Within the poor outcome predictions, the true poor-predicted ($n = 19$) group was characterized by genes associated with the extracellular matrix, and the false poor-predicted ($n = 30$) category was enriched for mitochondrial- and ribosomal-associated genes. ClusterProfiler[58] was used for both GO analysis with a Benjamini–Hochberg procedure.

extracellular matrix–associated genes, while the "false poor-predicted" group was enriched with ribosomal- and mitochondrial-associated genes (Fig. 7b).

Lastly, DEG and GO analyses for the "true favorable" ($n = 165$) and "true poor" ($n = 44$) groups were conducted (Supplementary Figs. 7c, 8, and Supplementary Data 3). Interestingly, immune response–related genes, including *PRSS16*, were enriched for the "true favorable" group, and extracellular matrix–associated genes for the "true poor" group, suggesting that PathoRiCH effectively captured the key features associated with both groups.

### *BRCA* mutation and HRD status prediction

To predict *BRCA* mutation status, MIL models were trained with the SEV and SMC cohorts ($n = 348$, WSI $n = 670$) and validated with the TCGA cohort ($n = 284$, WSI $n = 516$). The cancer-segmented area 20× magnification model showed the highest AUC-ROC value of 0.526 (Supplementary Table 7). HRD status was predicted based on three currently published HRD algorithms and using only the TCGA cohort ($n = 284$, WSI $n = 516$)[33–35]. With the HRD status algorithm by Telli et al., the cancer-segmented area 5× model showed the best performance, with an AUC-ROC of 0.524 (Supplementary Table 8). With the HRD status algorithm by Takaya et al., the cancer-segmented area 20× model showed the best performance, with an AUC-ROC of 0.556. However, none of the models could be successfully trained to determine HRD status using the algorithm by Perez-Villatoro et al., probably due to the small number of evaluated cases (83/284, 29.2%).

## Discussion

In this study, we developed the PathoRiCH classifier to predict PFI using histologic images alone. Among previous studies to develop prediction models for HGSOC using histological features (Supplementary Table 9)[22–26], only Yu et al. and Laury et al. aimed to predict PFI. Although

they achieved good performance using a convolutional neural network (CNN) architecture, they had limitations such as a heterogeneous study population (including low-grade serous carcinoma for Yu et al.), small study cohorts (training cohort $n = 30$ for Laury et al.), and a lack of external validation (both). In contrast, PathoRiCH uses a dual-stream (DS)-MIL, which is better suited than a CNN for discovering predictive morphologies from weakly labeled WSIs and is more effective than conventional MILs in addressing misclassification and overfitting. In addition, our model was trained with the well-curated cohort and validated with two external datasets (TCGA and SMC).

To identify the optimal model without overfitting to the internal cohort, we compared the performances of six trained models in the external (TCGA) cohort. Whereas the 5× models with both the all-tissue and cancer-segmented areas demonstrated the highest performance in the internal validation cohort, their performance decreased in the external validation cohort, where the 20× model showed the best results. This suggests that the cytologic features visible in 20× images are more crucial for predicting therapeutic responsiveness in HGSOC than the architectural features seen in 5× images. In addition, the superior performance of the 5× model in the internal cohort can be attributed to overfitting on low-resolution images. Notably, our study reveals that the multiscale models exhibited intermediate performance, falling between the 5× and 20× models. This consistent pattern persisted in a comparison with ensemble models of both 5× and 20× images. Although multiscale learning is generally expected to enhance performance by leveraging the strengths of the single-scale models, it can result in performance degradation due to various factors such as model interactions, data characteristics, or prediction targets[36,37]. In the context of predicting therapeutic responsiveness in HGSOC, the 20× single-scale model proved to be the best one.

The PathoRiCH-based classification was not associated with *BRCA* mutations or HRD status, as various other factors might affect the

response toward platinum treatment beyond homologous recombination mechanisms. Because PathoRiCH and the molecular traits were not associated, they could be synergistic when combined. Thus, we classified patients with HGSOC into four subgroups using the PathoRiCH+*BRCA* + HRD combination, and that classification provided higher predictive power than molecular biomarkers. Although platinum-based regimens have been standard primary systemic therapies for all patients with HGSOC, the favorable-*BRCA*/HRD-positive group was highly platinum-sensitive, suggesting that PARP inhibitors early on and extended surveillance frequency could be considered for this group. In contrast, the PathoRiCH-predicted poor group would be highly platinum-resistant, so promptly enrolling these patients in clinical trials for salvage treatments could be beneficial. Furthermore, PathoRiCH showed predictive power superior to that of current molecular biomarkers in both our survival and Cox regression analyses. Compared with molecular biomarkers, PathoRiCH is also a cost-effective solution because it requires only WSI images and does not necessitate additional tissue tests. Thus, it can readily be integrated into initial pathological diagnostic practice to provide risk stratification for patients.

Our visualization analysis revealed that "intratumoral lymphocytic infiltration" was a distinct histologic feature in the favorable group, and "hyperchromatic nuclei with poor cohesiveness and spindling features" and "cytoplasmic vacuoles and microcystic change" were distinct features in the poor group. Marked tumor infiltrating lymphocytes (TILs) are a well-studied feature associated with a favorable prognosis and response to platinum therapy in various cancer types[38–42]. On the other hand, reduced cell-to-cell cohesion and spindling morphologies are characteristic features of the epithelial-mesenchymal transition (EMT), which is known to be a factor contributing to resistance to platinum-based chemotherapy[43,44]. Interestingly, the histologic features identified for each group also correlated with our transcriptomic analysis: The correctly predicted favorable group showed enrichment of immune response–related pathways, and the correctly predicted poor group was enriched in extracellular matrix–associated pathways, which are associated with the EMT. The TIL and EMT traits have previously been reported to be associated with the platinum treatment response, but no standardized criteria have been established, and the subjectivity of their morphological assessment has hindered their integration into pathologic diagnoses. In this context, PathoRiCH innovatively adopted them for morphological classification. In addition, PathoRiCH's recognition of known prognostic features throughout its end-to-end learning process enhances the reliability of its decisions. On the other hand, "cytoplasmic vacuoles and microcystic change" is a poor group–specific histologic feature that has rarely been reported in malignancies[45–47]. Because its clinical significance is unclear, the biological origins of this histological trait need to be unraveled, which will require the advanced application of spatial transcriptomics in upcoming studies.

This study has limitations. First, PathoRiCH showed an AUC-ROC near 0.6 for the external validation cohorts, indicating that the model barely differentiated the two groups. This demonstrates the challenge of classifying HGSOC based solely on histological images, which arises in part because HGSOC is already histologically classified as high-grade. To ascertain the robustness of our model and introduce it into clinical practice, additional multicenter validations and in-depth interpretations of the models are essential. Second, regarding the confusion matrix of the ground truth and predicted response groups in the external (TCGA) cohort, a considerable proportion of the false poor-predicted cases were identified, which likely resulted from the internal training dataset with biased favorable ground truth group. To address this imbalance, more cases with poor response should be incorporated in the training dataset to balance the two groups and better train features associated with poor treatment response. Third, the TCGA cohort did not contain information on PARP inhibitor

administration. However, the clinical data for the TCGA cohort were collected only until 2010, and PARP inhibitors were not FDA-approved and introduced to ovarian cancer treatment until 2014[48]. Thus, the TCGA cohort is expected to be PARP inhibitor–naïve. Fourth, the transcriptomic analysis used bulk RNA sequencing data, which limited that analysis. To correlate the histological and molecular features of the PathoRiCH-predicted groups, we are conducting spatial transcriptomics for a future study. Lastly, our model showed suboptimal performance in predicting *BRCA* mutation and HRD status. As our training cohort contained only a small number of patients had *BRCA* mutation and HRD status data, large *BRCA*- and HRD-focused cohorts could be beneficial for improving our model.

In conclusion, we developed a histopathological image–based deep learning model with which to predict PFI in HGSOC. This model showed statistically significant performance in stratifying patients by PFI in three independent cohorts with different sample preparations and staining conditions. In addition, combining PathoRiCH with current molecular biomarkers provides an even more powerful tool for stratifying the risks of patients. The morphological and genetic features of the PathoRiCH-predicted groups support the high reliability of the model decisions. PathoRiCH does not require additional tissue tests or annotations from pathologists, allowing it to be straightforwardly implemented in clinical diagnostic practice. Our concept has the potential to transform the current diagnostic pipeline for HGSOC and guide gynecologic oncologists in selecting primary and maintenance treatments, planning surveillance frequency, and counseling patients about clinical trials.

## Methods
### Cohorts
This study was approved by the institutional review board of Severance Hospital (IRB no. 4-2021-1391). Informed consent was waived for this retrospective study and participants were not compensated. All patients were confirmed as female. For the SEV cohort, data for 394 patients with HGSOC (WSI, $n = 754$) between November 2005 and June 2022 were retrieved from Yonsei Severance Hospital (Seoul, Korea). The patients were stage III–IV at presentation, underwent primary debulking surgery followed by at least six cycles of adjuvant platinum-based chemotherapy, and did not receive PARP inhibitors in the first two years after diagnosis.

Of the 585 ovarian serous cystadenocarcinoma patients in the TCGA-OV dataset, those with *TP53* mutations, clinical outcome data including PFI, and H&E WSI were selected for the TCGA cohort (https://portal.gdc.cancer.gov/). After excluding cases with unevaluable slide images, 284 patients (WSI $n = 516$) with stage I–III HGSOC who underwent primary debulking surgery and platinum-based adjuvant chemotherapy were included in the analyses[49].

For the SMC cohort, data for 136 HGSOC patients (WSI, $n = 136$) treated between January 2018 and November 2021 were retrieved from Samsung Medical Center (Seoul, Korea). The patients were stage I–IV at presentation, underwent primary debulking surgery followed by at least six cycles of adjuvant platinum-based chemotherapy, and did not receive PARP inhibitors in the first two years after diagnosis. Clinicopathological data for the SEV and SMC cohorts were obtained from medical records and pathology reports.

### PFI, *BRCA* mutation, and HRD status prediction
For PFI prediction, patients were classified into binary response groups, with "platinum-resistant" (PFI ≤ 6 months) and "partially platinum-sensitive" (PFI 6–12 months) patients as the poor response group, and "platinum-sensitive" (PFI 12–24 months) and "very platinum-sensitive" (PFI > 24 months) patients as the favorable response group[27,28]. For training and internal validation, 5-fold cross-validation was performed in the SEV cohort ($n = 394$), and external validation was performed in the independent TCGA ($n = 284$) and SMC ($n = 136$)

cohorts. For *BRCA* mutation prediction, we trained the MIL models with a mixture of patients from the SEV and SMC cohorts who had available *BRCA* mutation status data (*n* = 348) and validated them with the TCGA cohort (*n* = 284). The SEV and SMC datasets do not contain HRD results; only the TCGA cohort (*n* = 284) was used for training and validating HRD status, with an 8:2 split for training and validation. The HRD status prediction was performed with three different HRD algorithms[33–35].

## Slide preparation and preprocessing

For the SEV and SMC cohorts, all H&E-stained slides were prepared from surgically resected specimens using formalin-fixed paraffin-embedded (FFPE) tissue blocks. WT-1 and p53 immunohistochemical staining was performed for the initial pathological diagnoses in all cases. All WSIs were retrospectively reviewed by gynecologic pathologists (E. Park, N.H. Cho, and H.-S. Kim), and the most representative slides with high tumor cellularity were selected, with a median number of 1.7 (range 1–5) slides per patient for the SEV cohort and 1 slide per patient for the SMC cohort. All slides were scanned using an Aperio AT2 scanner (Leica Biosystems, Wetzlar, Germany) at 20× magnification (0.50 μm/pixel). For the TCGA cohort, all H&E slide images were from FFPE or fresh-frozen surgical resection specimens before systemic treatment. We removed slides with no identifiable tumor tissue, low-resolution images, or artifacts such as large pen marks, tissue folds, or blurring.

WSIs are generally too large to be used as inputs for deep learning models, so the images must be divided into tiles. Tiles with less than 30% tissue area were filtered out to extract only tissue-containing tiles from the WSIs. Two types of tiles were created: tiles divided into 224 × 224 pixels to generate 20× tiles, and tiles divided into 896 × 896 pixels and resized to 224 × 224 pixels to generate 5× tiles. For the multiscale MIL model, the 5× and 20× tiles were constructed hierarchically.

## Model development

Three processes were applied sequentially to develop the MIL prediction models (Fig. 1): automated cancer segmentation for the cancer-segmented area models, contrastive self-supervised learning, and MIL modeling. All models were trained on a single NVIDIA RTX A6000 48GB GPU.

First, to automatically extract the cancer-segmented regions from the HGSOC WSIs, we developed a UNetPlusPlus-based cancer-segmentation model using ResNet34 as an encoder. In UNetPlusPlus, the encoders and decoders are connected through a nested dense skip path, making it a powerful architecture for medical image segmentation. The model was trained with 5× magnification patch images (448 × 448 μm with 224 × 224 pixels) generated from 810 breast invasive ductal carcinoma biopsy slides (tile *n* = 8130) from multiple institutions. In all the trained patch images, cell-level annotations for carcinomas were performed by experienced pathologists. After excluding patch images with a cancer area of <10%, 8130 patch images were created. The model was trained for 100 epochs using the SGD optimizer with weight decay and the Dice loss function. In the HGSOC WSIs, the model automatically distinguished cancerous areas from non-cancerous areas (Supplementary Fig. 1). To quantitively assess the performance of cancer segmentation in the HGSOC WSIs, we randomly selected 10% of cases from the SEV and TCGA cohorts (*n* = 39 and *n* = 28, respectively), and 250 × 250 μm regions were manually annotated by experienced pathologists.

Second, we used SimCLR self-supervised ResNet18 CNN models to visualize the 5× and 20× histological images[50–52]. The SimCLR algorithm allows the CNN to learn the representations of images by maximizing agreement among different augmented views of the same data example. The models were trained using the CAMELYON 16 and CAMELYON 17 datasets[53].

Third, we applied a DS-MIL for the prediction task[52]. DS-MIL provides improved classification and localization accuracy for multiscale WSI features by using a pyramidal fusion mechanism. For the MIL, a contrastive self-supervised CNN model was used as an extractor of histological features to produce powerful representations. The MIL model was trained for 200 epochs using the AdamW optimizer with weight decay, a cosine annealing learning rate, and a cross-entropy loss function. In contrast to similar previous studies that predicted and analyzed only one class as a binary classification, our MIL model predicted the probability of favorable and poor groups, respectively, to more precisely and clearly analyze the histological differences between the favorable and poor groups using attention. To define the final group based on the favorable and poor group prediction probabilities of the MIL model, we set an optimal threshold using the Youden's Index, which could find the point that maximizes the difference between the true positive rate and the false positive rate. The threshold for the favorable group was set to 0.623, and for the poor group, it was set to 0.377 (Supplementary Fig. 9). Consequently, any instance with a favorable prediction probability exceeding 0.623 was classified as belonging to the favorable group. Similarly, if the poor prediction probability surpassed 0.377, the instance was categorized as belonging to the poor group. If both the predicted probability for the favorable group and the predicted probability for the poor group were below or above those thresholds, the patient was placed in the prediction probability group with the maximum value. Each AUC-ROC was calculated using average values.

The accessible links for the utilized open-source versions are as follows. The program was developed using Python programming language (version 3.8). The models are implemented using PyTorch v1.10 (https://github.com/pytorch/pytorch) and Scikit-learn v1.0.2 (https://github.com/scikit-learn/scikit-learn/blob/main/sklearn/model_selection/split.py). The multi-instance learning process is based on DSMIL. The contrastive learning for feature extraction process is based on SimCLR (available at https://github.com/sthalles/SimCLR), with a ResNet18 backbone architecture (https://github.com/pytorch/vision/blob/master/torchvision/models/resnet.py). The cancer segmentation model is based on an implementation of UNetPlusPlus using segmentation-models-pytorch v0.3.3 (https://github.com/qubvel/segmentation_models.pytorch), with a ResNet34 backbone architecture (https://github.com/pytorch/vision/blob/master/torchvision/models/resnet.py). Mathematical operations are implemented using Numpy v1.23.4 (https://github.com/numpy/numpy). The WSIs were processed using Openslide-Python v1.3.0 (https://github.com/openslide/openslide-python). The patch image handler is based on OpenCV-Python v4.7.0.68 (https://github.com/opencv/opencv-python). Finally, dimension reduction and clustering process is implemented using Scikit-Learn (https://github.com/scikit-learn/scikit-learn/tree/main/sklearn/decomposition and https://github.com/scikit-learn/scikit-learn/tree/main/sklearn/mixture).

## Patient level classification

In the slide-level evaluation, patients with multiple slides could be predicted as belonging to different groups according to the evaluated slides. In such cases, the procedure entailed computing the average favorable prediction probability and average poor prediction probability for each patient. The subsequent steps followed the same methodology for final group definition used for the slide-level prediction.

## Ensemble analysis

We proceeded with an ensemble technique, aggregating results of the 5× and the 20× MIL models through either soft or hard voting, based on a previous study[54]. Soft voting determines the final prediction class by averaging prediction probabilities across models and applying a threshold, whereas hard voting relies on the proportion of positive

predictions among classes. Our hard voting method includes both the AND and OR conditions. Under the AND condition, the final prediction is labeled as the poor group only if both models predict the poor group. Conversely, with the OR condition, the final prediction is assigned to the poor group if either model predicts it. The threshold for the soft voting ensemble model was set using the Youden's Index, with 0.533 for favorable group and 0.467 for poor group.

### High-score patch extraction and clustering analysis
Each patch image was represented as an one-dimensional feature vector using the global average pooling output of the final convolutional layer of the contrastive self-supervised CNN. We obtained the maximum and minimum values from all patch attention scores, and the normalized attention scores were obtained by applying min-max normalization.

After dimensionally reducing the one-dimensional feature vectors using the t-SNE algorithm, the GMM algorithm was used for clustering. The initial numbers of GMM clusters were set according to the heuristic and default parameters of the GMM and t-SNE, respectively.

### TCGA transcriptome analysis
All analyses on TCGA data were performed using R (v 4.2.1). The 421 RNAseq STAR counts data from primary tumor specimens from the TCGA-OV project were downloaded and processed using TCGAbiolinks (version 2.31.1)[55]. Of the 284 TCGA patients used with PathoRiCH, 208 patients with available RNAseq results were analyzed. Insufficiently expressed genes were filtered by the default filterByExpr function, and expression counts were normalized to Trimmed Mean of M-values (TMM) by the calcNormFactors function in edgeR (version 3.38.4)[56]. TMM data were transformed to log2-counts per million using the voom function in limma (version 3.52.4)[57]. Model fitting and extracting DEGs were performed by the lmFit, eBayes, and topTable functions in limma. A volcano plot of the differential expression fold change data was created with cut-off of absolute log2 fold change >1 and $p < 0.01$, using the EnhancedVolcano package (version 1.14.0). A gene set enrichment analysis comparing the two response groups was performed using the gseGO function of clusterProfiler (version 4.4.4)[58], with a Benjamini–Hochberg procedure for multiple testing correction.

### BRCA genetic tests
In SEV and SMC cohorts, BRCA mutation status was assessed in peripheral blood and tumor samples. Germline BRCA was tested using Sanger sequencing or next-generation sequencing (NGS). Sanger sequencing was performed on a 3730 DNA Analyzer with a BigDye Terminator v3.1 Cycle Sequencing Kit (Applied Biosystems, Foster City, CA, USA), followed by analysis using Sequencher 5.3 software. NGS using a custom panel, including BRCA1 and BRCA2 genes, was performed in a proportion of patients on a NextSeq 550 instrument (Illumina) with 2 × 151 bp reads. Bioinformatic analysis was performed using the Burrows-Wheeler Aligner, Genome Analysis Toolkit, Ensembl Variant Effect Predictor, and a custom pipeline. Experienced geneticists made final interpretations. For tumor BRCA, genomic DNA was extracted for NGS of tumor samples using a Maxwell CSC DNA FFPE Kit (Promega, Madison, WI, USA) according to the manufacturer's instructions. The products were sequenced using the Nestxeq550 System (Illumina) using the TruSight Oncology 500 panel (Illumina). For mutational analysis, FASTQ files were uploaded to the Illumina BaseSpace software (Illumina) for variant interpretation. Only variants in the coding regions, promoter regions, or splice variants with a minimum 3% of the reads and read depth of 250.

### Statistics and reproducibility
No statistical method was used to predetermine sample size and all available samples were included in the model development and validations. No data were excluded from the analyses and the analyses were not randomized. The researchers were blinded to the labels of the samples in the test set before the final model evaluation.

A $\chi^2$ test was used to evaluate correlations between categorical variables. Kaplan–Meier survival curves were estimated in the survival analysis. Two-sided unpaired $t$-tests, Mann–Whitney $U$ tests, and Wilcoxon rank-sum tests were used to compare differences. A $p < 0.05$ was considered statistically significant for all analyses, and all analyses were two-tailed. All data were analyzed using R software (version 4.2.1).

### Reporting summary
Further information on research design is available in the Nature Portfolio Reporting Summary linked to this article.

## Data availability
Source data are provided as a Source Data file. The remaining data are available within the Article or Supplementary Data. The TCGA dataset is publicly available via the TCGA portal (https://portal.gdc.cancer. gov). The CAMELYON 16 and CAMELYON17 dataset is publicly available via Camelyon grand challenge website (https://camelyon16.grand-challenge.org, and https://camelyon17.grand-challenge.org respectively). SEV and SMC cohorts' NGS dataset for germline BRCA tests is deposited under accession number PRJNA1108881. WSI data for the SEV and SMC cohorts are not publicly available due to hospital regulations. The data could be available on request from the corresponding author (E.P.) and response will be received typically within 4 weeks. Data usage is restricted to non-commercial academic research purposes. Source data are provided with this paper.

## Code availability
The source code is publicly available and can be downloaded from https://github.com/dmmoon/PathoRICH. Additional requests or inquiries about the code can be made to dmmoon@jlkgroup.com.

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

## Acknowledgements

This work was supported by a National Research Foundation of Korea (NRF) grant funded by the Korean government (MSIT) (Grant No. 2021R1G1A1091264 to E.P. and 2023R1A2C2006223 to H.-S.K.), a faculty research grant from Yonsei University College of Medicine (6-2021-0244) to E.P., and grants from the Korean Health Technology R&D Project through the Korean Health Industry Development Institute (KHIDI), funded by the Ministry of Health and Welfare, Republic of Korea (Grant No: HC21C0012 to E.P. and HI21C0977 to N.H.C.). The authors thank Seo Hee Kim and So Young Choi for scanning the slides.

## Author contributions

Conceptualization: N.H.C., H.-K.C., D.M.K., J.-Y.L., and E.P.; Methodology: D.M., N.H.C., H.-K.C., and D.M.K.; Software: D.M., H.-K.C., and D.M.K.; Validation: B.A., D.M., H.-K.C., D.M.K., and E.P.; Formal Analysis: B.A., D.M., C.L., H.-K.C., D.M.K., and E.P.; Investigation: B.A., C.L., and E.P.; Resources: H.-S. K., J.-Y.L., E.J.N., D.W., H.-J.A., S.Y.K., S.-J.S., H.R.J., D.K., H.J.P., M.K., Y.J.C., H.P., Y.L., S.N., Y.-M.L., S.E.C., J.K., S.H.S., and E.P.; Data Curation: B.A., H.-J.A., S.Y.K., S.-J.S., H.R.J., D.K., H.J.P., M.K., Y.J.C., H.P., Y.L., S.N., Y.-M.L., S.E.C., J.K., S.H.S., and E.P.; Writing–original draft: B.A., D.M., C.L., H.-K.C., D.M.K., D.W., and E.P.; Writing–review & editing: B.A., D.M., C.L., H.-K.C., D.M.K., H.-S.K., and E.P.; Visualization: B.A., D.M., C.L., H.-K.C., D.M.K., and E.P.; Supervision: N.H.C., H.-K.C., D.M.K., J.-Y.L., E.J.N., H.-J.A., H.-S.K., and E.P.; Project Administration: N.H.C., H.-K.C., D.M.K., and E.P.; Funding Acquisition: N.H.C., D.M.K., and E.P. All the authors critically read, edited, and approved the final manuscript.

## Competing interests

The authors declare no competing interests.

## Additional information

¹Department of Pathology, Severance Hospital, Yonsei University College of Medicine, Seoul, South Korea. ²Artificial Intelligence Research Center, JLK Inc., Seoul, South Korea. ³Department of Pathology and Translational Genomics, Samsung Medical Center, Sungkyunkwan University School of Medicine, Seoul, South Korea. ⁴Department of Obstetrics and Gynecology, Institute of Women's Life Medical Science, Yonsei University College of Medicine, Seoul, South Korea. ⁵Department of Laboratory Medicine, Yonsei University College of Medicine, Seoul, South Korea. ⁶Department of Pathology, CHA Bundang Medical Center, CHA University School of Medicine, Seongnam, South Korea. ⁷Department of Pathology, Keimyung University School of Medicine, Daegu, South Korea. ⁸Department of Pathology, Gangnam Severance Hospital, Yonsei University College of Medicine, Seoul, South Korea. ⁹Institute of Breast Cancer Precision Medicine, Yonsei University College of Medicine, Seoul, South Korea. ¹⁰Department of Diagnostic Pathology, Gangnam CHA Medical Center, CHA University College of Medicine, Seoul, South Korea. ¹¹Department of Pathology, Dankook University School of Medicine, Cheonan, South Korea. ¹²Department of Pathology, Ewha Womans University, Seoul, South Korea. ¹³These authors contributed equally: Byungsoo Ahn, Damin Moon, Hyun-Soo Kim. ✉e-mail: epark54@yuhs.ac

