## [Peer Review File · Nature Communications]

REVIEWER COMMENTS

Reviewer #1 (Remarks to the Author): Expert in digital pathology, deep learning, computational genomics, and ovarian cancer

The authors present a framework to predict response to adjuvant chemotherapy of high-grade serous ovarian cancer patients. This is an important topic and applications of computational pathology to ovarian cancer are still relatively limited. The authors analyze two datasets including an internal dataset and a TCGA dataset, adding up to several hundred patients. The paper is well written and easy to read, however I have some concerns regarding the methodology, the performance of the models and the impact of the results.

Major comments

- 1) The major limitations of previous studies are said to be the lack of external validation, dataset size, and model reliability. This manuscript addresses the first point, but not the other two. Dataset size seems to limit performance, and the models do not perform consistently in the internal and external cohorts, which suggests that their performance assessment is not reliable.
- 2) What was the performance of the cancer detection model on the internal and external validation sets? This is important to understand to what extent it's biasing the overall model performance.
- 3) What was the performance of the cell classification model on the internal and external validation sets? If the performance is decent, could this help to improve the identification of cancer regions? If it's not, is it biasing the downstream cell type analysis?
- 4) Given how dataset size and overfitting seems to be an important limitation, could you train the models using cross-validation instead of simple validation? The models were trained on a single GPU which suggests that they were not extremely computationally expensive.
- 5) The whole purpose of the internal validation set is to select the best performing model, to then further confirm its generalisability in the external set. Testing all models on the external set and then choosing the best one limits the value of the whole validation strategy, because you are essentially using the external validation set to "fine-tune" your model selection. To truly demonstrate the performance of the model you would now need another dataset.
- 6) The performance of the classification models is somewhat hidden in the results section. Please add a table with all the AUC results and any other relevant performance score.
- 7) Why is pooling the two magnifications worse than the individual ones independently? Did you try different ensembling strategies?
- 8) I have not been able to fully understand the PFI analysis. How exactly does the training work? Are two models trained, one for "favourable" and one for "poor" subgroups? The binarisation that is performed in the Kaplan Meier curves, is it based on the predicted scores? If the answer is yes, how exactly does that work, do you cut at 0.5, or... ?
- 9) The transcriptomic analysis does not add much in the way it is defined: the value would be in doing a differential analysis between the transcriptomic profiles of "real" good responders vs. "predicted" good responders, to understand which features of the ones that are being identified by the model in the histology are truly revealing underlying molecular biology.

Minor comments

- 1) Please include number of datasets and dataset size in the abstract.
- 2) line 50: "an innovative tool that would transform the current diagnostic pipeline for HGSOC and improve patient outcomes" is a bold claim given the relatively poor performance of the model. These results prove the feasibility of the idea but would need significant improvement in performance to be applicable and have impact clinically.
- 3) line 78: "lacks histological grading" -> what does this mean exactly? HGSOC is high grade by definition
- 4) Please define which dataset is "internal" and which "external" (assuming TCGA is "external" but it would be good to define it explicitly.)
- 5) line 299: this should be "proves" instead of "disproves" I believe?
- 6) line 300: I would refer to "statistically significant results" rather than "remarkable performance"

Reviewer #2 (Remarks to the Author): Expert in ovarian cancer genomics, imaging, and therapy

The authors introduce a novel deep-learning approach for predicting patient responses to platinum-based chemotherapy. Although similar studies exist (highlighted by authors in the paper), the researchers distinguish their work through the use of UNetPlusPlus model, external validation, and insights into the model's decision-making process. While the paper has value in bringing an important improvement into the field, the results are not outstanding in terms of prediction (AUC-ROC best values 0.7). Moreover, the scientific novelty is limited to method development, and the study provides little new insights into the biology of ovarian cancer. The addition of other data levels, e.g. spatial transcriptomics on the high-score patches, or highly multiplexed staining of the tissues to confirm the bulk RNAseq deconvolution analyses would significantly increase the biological and clinical impacts of the study.

Key Points:

- Choice of Analysis Groups (Lines 113-114): The rationale for selecting two analysis groups, favourable (≤ 12 months) and poor prognosis (> 24 months), lacks explicit mention. The study should transparently clarify this choice based on clinical relevance and existing knowledge.
- Cancer Segmentation Model Validation (Breast Invasive Ductal Carcinoma) (Lines 120-123): Due to using a model trained on a different cancer type (breast invasive ductal carcinoma) for cancer segmentation, validating its performance on HGSOC samples is crucial. This step is pivotal since the study eventually opts for the cancer-segmented area at 20 \times magnification for MIL.
- Calculation of Predictive Value per Slide (Lines 169-170): The calculation method for predictive value per slide is not explicitly explained. Elaborating on this calculation is important for comprehending its significance, especially as it is used to predict patients with multiple slides.
- The selection criteria for HRD: The authors should explain why they have used Takaya et al algorithm

for HRD classification. It would be important to show whether the use of improved and clinically used algorithms (PMID 26957554, 36581696) or detection of mutational signatures (PMID 30988514) might improve the separation of the HRD groups, and thus the performance of the prediction.

Minor Issues:

- Performance Comparison of PathoRiCH (Lines 281-282): While the sentence highlights PathoRiCH's superior performance, it's important to note that Supplementary Table 5 displays models with higher AUC values. Providing context on the specific metric used to assess its superior performance, or emphasizing the model's extensive training and validation on a larger dataset, could improve this statement.
- "Additional Tissues" (Line 73): The term "additional tissues" requires clarification. It could be helpful to elaborate on whether this refers to a larger sample size or a different concept.
- Supplementary Material Mistakes (Line 298): Typos, like the one found ("insitution"), should be rectified in the supplementary material.

Additional Considerations

- Mitigating Impact of Random Splits: The consistency of performance metrics between external and internal validation of the prediction model demonstrates limited overfitting to the training set. However, it is still important to consider the inherent variability in performance metrics due to the random nature of data in model training, as this variability introduced by chance can impact results. To ensure robust model evaluation, techniques such as cross-validation or repeated experimentation could be considered to minimize the influence of random splits and provide a more stable estimation of model performance.
- Comparative Analysis of Initial Data Distribution and Predictions (Lines 170-175): It might be insightful to compare the initial distribution of data in Kaplan-Meier survival analysis plots for the initial favourable and poor prognosis groups and the data distribution depicted by the predicted groups' figures. Such a comparison could aid in assessing the model's performance and identifying areas that require refinement.
- TCGA Cohort and PARP Inhibitor Information (Lines 110-111): The absence of PARP inhibitor administration information in the TCGA cohort could limit the study's findings. External validation results could be influenced by the unavailability of this data, affecting the model's generalizability to real-world scenarios involving PARP inhibitors.

Reviewer #3 (Remarks to the Author): Clinical expert in ovarian cancer and pathology

This study addresses risk stratification of HGSOC using a combination of AI based image analysis with

BRCA/HRD status to predict response to chemotherapy. The results are very promising in showing the predictive value of this combined approach for patient prognosis. This addresses a yet unmet clinical need and has the potential of being translated to other types of cancer.

The choice of study cohorts and the division between training and validation sets seems reasonable. The authors have also provided sufficient details in the methodology to help in attempts to reproduce the work or inform other researchers in the field for comparison with other methods. However, the numbers of cases studied are still limited to ascertain the robustness of the system and illegibility of being rolled out to routine practice. This needs to be well highlighted in the discussion and prospective plans to address that suggested to help in planning future studies that build on this work.

Manuscript ID: *NCOMMS-23-33128*

Title: “Histopathologic Image-Based Deep Learning Classifier for Predicting Platinum-based Treatment Response in High-Grade Serous Ovarian Cancer”

We appreciate the referees' suggestions that motivated us to improve the quality of this work, and have carefully revised the manuscript in the light of the referees' advice. Our point-by-point responses to the reviewers' comments are provided below. Please find hereafter our answers (in **blue**) to the reviewers' comments (in **black**); and for convenience, the changed parts and newly added parts for the revised manuscript have been highlighted in a yellow background.

The main improvements of the revised manuscript are as below:

- 1) **Another new external validation cohort:** We introduced an additional external cohort for model validation, comprising 136 patients from Samsung Medical Center in Seoul, Korea.
- 2) **Augmentation of the size of training and internal (SEV) validation cohort:** We added 156 patients with fulfilling inclusion criteria to the SEV cohort, bringing the total up to 394 patients.
- 3) **Implementation of 5-fold cross-validation for the model:** To enhance model reliability, we re-developed our approach by implementing 5-fold cross-validation methods.
- 4) **Revision of the cut-off for platinum-free interval:** According to the clinical relevance, we used a 12-month cut-off for defining favorable and poor response groups.
- 5) **Transcriptomic analysis:** We compared the three combinations of subgroups for analysis; “true favorable-predicted”–“false favorable-predicted” groups, “true favorable-predicted”–“false favorable-predicted” groups, and “true poor-predicted”–“false poor-

predicted” groups.

The minor changes, not described in the “Response to the Reviewers’ Comments,” are as below:

- 1) **Detailed results and methods:** According to the revised prediction model through the new training cohort and validation methodology, all subsequent analyses were re-evaluated. In addition, the results are more clearly presented (ex. K-M analysis results for less important models are presented as K-M p values in the table, rather than survival graphs).
- 2) **Removal of “initial model” part:** Considering relevance to main contents and the length limit of manuscript, we excluded the preliminary results using convolutional neural network-based model, which were demonstrated in the Results–“initial model” part (lines 119–139 in the original manuscript).
- 3) **Patch selection for cluster analysis:** For a more accurate analysis, an analysis was performed using equal number ($n = 3,500$) of high score patches from favorable and poor groups.

Response to the Reviewers' Comments

Reviewer #1 (Remarks to the Author):

The authors present a framework to predict response to adjuvant chemotherapy of high-grade serous ovarian cancer patients. This is an important topic and applications of computational pathology to ovarian cancer are still relatively limited. The authors analyze two datasets including an internal dataset and a TCGA dataset, adding up to several hundred patients. The paper is well written and easy to read, however I have some concerns regarding the methodology, the performance of the models and the impact of the results.

Major comments

1) The major limitations of previous studies are said to be the lack of external validation, dataset size, and model reliability. This manuscript addresses the first point, but not the other two. Dataset size seems to limit performance, and the models do not perform consistently in the internal and external cohorts, which suggests that their performance assessment is not reliable.

Thank you for your comprehensive feedback. To bolster the issues about dataset size and model reliability, we have undertaken a comprehensive re-development and re-validation of our model. First, we augmented the size of our internal (SEV) cohort by incorporating an additional 156 patients, bringing up the total from 238 to 394. This addition establishes the current study as containing the largest training cohort size within the landscape of developing ovarian cancer prediction models.

With the augmented training cohort, the performances of the all-tissue area MIL models in the external (TCGA) cohort were increased, suggesting that the overfitting issue of all tissue area models has been considerably resolved (Table 1 below). The cancer-segmented area MILs showed similar AUC-ROC values compared to our original MIL models (Table 2 below);

however, the gap between the internal and external validation was also decreased. Regarding a balance of the performance evaluation metrics including AUC-ROC and F1-score and consistency in the internal and external cohorts, the cancer-segmented area 20× MIL still showed the best performance for predicting platinum-treatment response.

To enhance the reliability of our model, we re-developed our approach by implementing 5-fold cross-validation methods, ensuring that our results were not merely a product of chance. In addition, to further demonstrate the reliability, we introduced an additional external validation cohort, comprising 136 patients from Samsung Medical Center (Seoul, Korea). Our selected model displayed an AUC-ROC value of 0.593 and F1 score of 0.711, which were consistent with the results of the SEV and TCGA cohorts.

Table 1. Performance of multiple-instance learning models in the internal (SEV) and external (TCGA and SMC) validation cohorts in predicting platinum-based treatment response groups (shown as Table 1 in the revised manuscript)

		All-tissue area MIL			Cancer-segmented area MIL		
		5×	20×	Multiscale	5×	20×	Multiscale
Internal validation (SEV cohort)	AUC-ROC*	0.627 ± 0.047	0.610 ± 0.04	0.623 ± 0.016	0.604 ± 0.05	0.596 ± 0.072	0.614 ± 0.046
	Precision	0.495	0.605	0.565	0.521	0.465	0.507
	Recall	0.663	0.411	0.545	0.468	0.675	0.525
	F1 score	0.559	0.462	0.517	0.470	0.522	0.507
	K-M p value**	0.000	0.000	0.000	0.000	0.000	0.000
External validation (TCGA cohort)	AUC-ROC	0.492	0.594	0.575	0.532	0.602	0.573
	Precision	0.187	0.253	0.232	0.519	0.406	0.407
	Recall	0.879	0.484	0.429	0.250	0.528	0.481
	F1 score	0.309	0.332	0.301	0.338	0.459	0.441
	K-M p value**	0.108	0.004	0.000	0.000	0.032	0.036

External validation (SMC cohort)	AUC-ROC	-	-	-	-	0.593	-
	Precision	-	-	-	-	0.551	-
	Recall	-	-	-	-	0.351	-
	F1 score	-	-	-	-	0.711	-
	K-M p value**	-	-	-	-	0.030	-

AUC-ROC, area under the receiver operating characteristic curve; K-M, Kaplan-Meier analysis; MIL, multiple instance learning

*From 5-fold cross validation; **Based on platinum-free interval

Table 2. Performance of the original multiple instance learning models

Scale	All-tissue area MIL			Cancer-segmented area MIL		
	5×	20×	Multiscale	5×	20×	Multiscale
Internal validation						
AUC-ROC	0.698	0.711	0.664	0.649	0.615	0.681
External validation						
AUC-ROC	0.536	0.553	0.497	0.495	0.574	0.447

AUC-ROC, area under the receiver operating characteristic curve; MIL, multiple instance learning

2) What was the performance of the cancer detection model on the internal and external validation sets? This is important to understand to what extent it's biasing the overall model performance.

To validate our cancer segmentation model, we performed a concordance analysis between the pathologist-annotated area and the model's cancer segmented area. In detail, 10% of cases from the internal (SEV) and the external (TCGA) cohorts were randomly selected, and 250x250 μm regions from the whole slide images were manually annotated by experienced gynecologic

pathologists. Then, we compared the cancer segmented areas between pathologists and our cancer segmentation model.

As a result, our model demonstrated consistently high concordance across both the internal (SEV) and external (TCGA) cohorts, exhibiting mean Dice coefficients of 0.781 and 0.836, respectively (Table below). Upon reviewing the images of cancer segmented area by pathologists and our segmentation model, the model often missed small cancer areas (Figure below). However, a considerable disparity came from the treatment of interstitial and white background spaces between cancer cells, where the segmentation model displayed a more intricate exclusion of these spaces compared to human pathologists. In that regard, the actual accuracy of the cancer segmentation model might be higher than the calculated value. We included this analysis in the Results and Methods sections of our revised manuscript (page 20, line 412).

Table. Cancer segmentation concordance between pathologists and the cancer segmentation model (included as Supplementary Table 1 in the revised manuscript)

	Dice coefficient (Mean)
SEV cohort (n=39)	0.781
TCGA cohort (n=29)	0.836
Overall	0.804

Figure. The representative images of cancer segmentation between pathologists and the cancer segmentation model (included as Supplementary Figure 2 in the revised manuscript)

3) What was the performance of the cell classification model on the internal and external validation sets? If the performance is decent, could this help to improve the identification of cancer regions? If it's not, is it biasing the downstream cell type analysis?

For the four distinct cell types (epithelial cells, lymphocytes, plasma cells, and connective cells), we performed concordance analyses between the pathologist-annotated area and the model's

cancer segmented area, similar to the validation process for our cancer segmentation model. However, the concordance results were dismal: the mean Dice coefficients for each cell type were as follows: 0.413 for epithelial cells, 0.099 for lymphocytes, 0.029 for plasma cells, and 0.245 for connective cells (Table below). The model seemed to face difficulties due to the intricate and variable morphology of cell types, subtle variations of staining patterns, and overlapping cells. Accordingly, the cell quantification results of the original manuscript were not reliable. In our revised manuscript, we removed the related analysis, which were previously used for explaining our model decisions.

Table. Concordance of cell type identification between pathologists and the cell segmentation model

	Dice coefficient (Mean)			
	Epithelial cells	Lymphocytes	Plasma cells	Connective tissue cells
SEV cohort (n=39)	0.456	0.147	0.058	0.279
TCGA cohort (n=29)	0.370	0.051	0.000	0.211
Overall	0.413	0.099	0.029	0.245

4) Given how dataset size and overfitting seems to be an important limitation, could you train the models using cross-validation instead of simple validation? The models were trained on a single GPU which suggests that they were not extremely computationally expensive.

Taking your perspectives into account, we conducted a 5-fold cross-validation in our revised model. For internal validation, the final performance metrics were refined to present the average and standard deviation values across each fold. For external validation, we employed the best AUC-ROC model among the folds. The performance of 5-fold cross-validation (Table) is referred to our response to your major comment #1.

5) The whole purpose of the internal validation set is to select the best performing model, to then further confirm its generalisability in the external set. Testing all models on the external set and then choosing the best one limits the value of the whole validation strategy, because you are essentially using the external validation set to "fine-tune" your model selection. To truly demonstrate the performance of the model you would now need another dataset.

As you mentioned, we selected the model considering the performance in both the internal (SEV) and external (TCGA) validation cohorts, which could be a straightforward approach to overcome the model's overfitting issue.

As you requested, we have integrated an additional independent external dataset obtained from Samsung Medical Center (SMC, Seoul, Korea), consisting of 136 patients, for the sole purpose of assessing the performance of the model. While the AUC-ROC values for internal (SEV) and external (TCGA) cohorts were 0.596 ± 0.072 and 0.602, respectively, the model exhibited a consistent performance in the additional external (SMC) cohort, with an AUC-ROC value of 0.593 and F1 score of 0.711.

6) The performance of the classification models is somewhat hidden in the results section. Please add a table with all the AUC results and any other relevant performance score.

We added additional performance metrics, including precision, recall, and F1 scores, as well as AUC-ROC. We presented the Table as main Table 1 in the revised manuscript. The Table has also been referred in our response to your major comment #1.

7) Why is pooling the two magnifications worse than the individual ones independently? Did you try different ensembling strategies?

According to previous studies to predict cancer subtypes or molecular subtypes, multiscale MIL model had generally been demonstrated to have superior performance to single-scale models by leveraging the strengths of both 5× and 20× models¹⁻³. However, unexpectedly, our study revealed that the multiscale model exhibited intermediate performance, falling between the 5× and 20× models. Given that our task involves predicting therapeutic responsiveness in ovarian cancer, the results could differ from previous studies. Our study showed that the 5× model tends to overfit to our internal data, more so than the 20× model. Therefore, when we combined 5× and 20× information in the multiscale model, it ended up being overly influenced by the 5× data. As a result, the performance in checking against external data was not as good as the simpler 20× model on its own.

To provide additional context, the multiscale model was trained using data that involved the concatenation of feature vectors from both the 5× and 20× patch images. Specifically, employing the resnet18 model, a 512-dimensional vector encapsulating the features of the 5× patch image was combined with another 512-dimensional vector representing the features of the 20× patch image. This concatenation resulted in a unified vector of size 1,024, which was then utilized for training the multiscale model. This approach differed from conventional ensemble techniques.

Reflecting your comments, we updated the table to include the ensemble results of the 5× model and the 20× model, providing a comprehensive view. According to the literature references, we used an ensemble technique that combines results from two or more models through a voting method⁴. There are two types of voting methods: soft voting and hard voting. Soft voting determines the final prediction class by averaging the prediction probabilities of two or more models, and then applying a threshold. On the other hand, hard voting decides the final prediction class based on the proportion of positive predictions among two or more prediction classes from the models. In the hard voting method, we employed both the AND condition and the OR condition. Using the AND condition, the final prediction was categorized as the "poor" group only when both models predicted the "poor" group. Conversely, with the OR condition, the final prediction was assigned to the "poor" group if either of the models predicted the "poor" group.

Despite conducting an ensemble of results from both the 5× and 20× models, it has been validated that the individual outcome from the single 20× model stands out the most, particularly in terms of the F1 score (Table below). We included this analysis in the Results and Methods sections of our revised manuscript (page 21, line 446).

Table. Performance of ensemble analysis for the internal (SEV) and external (TCGA) validation cohorts (shown as Supplementary Table 2)

		Cancer-segmented area						
		5×	20×	Multiscale	Soft voting (5×, 20×)	Hard voting (AND) (5×, 20×)	Hard voting (OR) (5×, 20×)	
Internal validation	AUC-ROC*	0.604 ± 0.05	0.596 ± 0.072	0.614 ± 0.046	0.586	0.635	0.587	

	Precision	0.521	0.465	0.507	0.424	0.465	0.518
	Recall	0.468	0.675	0.525	0.562	0.739	0.374
	F1 score	0.470	0.522	0.507	0.483	0.566	0.418
	K-M p value**	0.000	0.000	0.000	0.059	0.011	0.149
External validation (TCGA cohort)	AUC-ROC	0.532	0.602	0.573	0.553	0.579	0.578
	Precision	0.519	0.406	0.407	0.500	0.398	0.556
	Recall	0.250	0.528	0.481	0.245	0.538	0.250
	F1 score	0.338	0.459	0.441	0.329	0.457	0.345
	K-M p value**	0.000	0.032	0.036	0.000	0.023	0.000

AUC-ROC, area under the receiver operating characteristic curve; K-M, Kaplan-Meier analysis

*From 5-fold cross validation; **Based on platinum-free interval

8) I have not been able to fully understand the PFI analysis. How exactly does the training work? Are two models trained, one for "favourable" and one for "poor" subgroups? The binarisation that is performed in the Kaplan Meier curves, is it based on the predicted scores? If the answer is yes, how exactly does that work, do you cut at 0.5, or... ?

For the first question, one MIL model learning process involves determining the likelihood that a given whole slide image (WSI) belongs to either the favorable or the poor group. This is achieved by calculating the probabilities for both the favorable and poor groups associated with the input WSI and minimizing the error between these predictions and the actual ground truth labels. For the second question, the binarization for the favorable or poor response groups is based on the predicted values of the MIL model with optimal thresholds. We set the threshold at 0.623 for the predictable favorable group and 0.377 for the predictable poor group. Accordingly, if the predicted score for the favorable group was higher than 0.623, it was classified into the favorable group; if the predicted score for the poor group was higher than 0.377, it was classified into the poor group. If both the predicted probability for the favorable

group and the predicted probability for the poor group were either below or above thresholds, the patient group was defined as the prediction probability group with the maximum value between the favorable prediction probability and the poor prediction probability. We chose the optimal thresholds by considering the trade-off between false positive rate and false negative rate, setting the optimal threshold at the intersection point of these rates. We included this detailed description of our model in the Methods section of our revised manuscript (page 20, line 428).

9) The transcriptomic analysis does not add much in the way it is defined: the value would be in doing a differential analysis between the transcriptomic profiles of "real" good responders vs. "predicted" good responders, to understand which features of the ones that are being identified by the model in the histology are truly revealing underlying molecular biology.

As you pointed out, to comprehend the essential histologic features predicted by the model, it is more appropriate to compare the histologic features that "true favorable/poor (the model has correctly learned)" with those "false favorable/poor (the model has misinterpreted)," rather than solely comparing the predicted "favorable" and "poor" groups. For the TCGA cohort comprising 208 individuals with available RNAseq results, the model's classification resulted in four distinct groups: true/false favorable-predicted groups and true/false poor-predicted groups (Table below).

We analyzed the different RNA expression patterns between the "true favorable-predicted" and "false favorable-predicted" cases. Similar comparisons were drawn between the "true poor-predicted" and "false poor-predicted" cases. Additionally, we compared the "true favorable" and "true poor" cases.

Table.

		Predicted		Total
		Favorable	Poor	
Ground truth	Favorable	True favorable-predicted 134 (84.4%)	False poor-predicted 30 (61.2%)	164
	Poor	False favorable-predicted 25 (15.6%)	True poor-predicted 19 (38.8%)	44
Total		159	49	208

As you rightly commented, our previous analysis turned out to have misinterpreted the model results. In gene ontology analysis, enrichment of immune response-related genes was observed in the “true favorable-predicted” group, while ribosomal and mitochondrial associated genes were enriched in the “false favorable-predicted” group (Fig 6a). In comparison of “true poor-predicted” and “false poor-predicted” groups, extracellular matrix-associated genes were enriched in the “true poor-predicted” group, while ribosomal and mitochondrial associated genes were enriched in the “false poor-predicted” group (Fig 6b). Interestingly, gene ontology analysis for the “true favorable” (n = 165) and “true poor” (n = 44) groups showed enrichment of immune response-related genes for the “true favorable” group and extracellular matrix-associated genes for the “true poor” group, suggesting PathoRiCH caught the key features of both groups (Supplementary Figure 7).

Minor comments

1) Please include number of datasets and dataset size in the abstract.

We appreciate your thoughtful feedback. In response to your comment, we have revised the

abstract to incorporate details about the number of cohorts and their respective sizes, including the newly added Samsung Medical Center cohort.

After revision (page 3, line 43): PathoRiCH was trained on an in-house cohort (n = 394) and validated on two independent external cohorts (n = 284 and n = 136).

2) line 50: "an innovative tool that would transform the current diagnostic pipeline for HGSOC and improve patient outcomes" is a bold claim given the relatively poor performance of the model. These results prove the feasibility of the idea but would need significant improvement in performance to be applicable and have impact clinically.

We agree with your opinion. We acknowledge the suboptimal AUC-ROC value, which currently rests at the 0.6 mark. The suboptimal performance of our model might be attributed to the inherent poor differentiation of high-grade serous ovarian carcinoma (HGSOC), considering HGSOC is already histologically classified as high-grade. According to your suggestion, we toned down the description as below.

Before revision: We believe that our method is an innovative tool that would transform the current diagnostic pipeline for HGSOC and improve patient outcomes.

After revision (page 3, line 50): PathoRiCH will provide a solid foundation for developing an innovative and reliable tool to transform the current diagnostic pipeline for HGSOC.

However, in terms of patient stratification, our model exhibited superior prognostic performance compared to molecular biomarkers, which are currently regarded as the most powerful predictors. Additionally, the combination of the histologic (PathoRiCH) and molecular (BRCA and HRD status) biomarkers exhibited more detailed patient stratification. This highlights the predictive potential of tumor histology in HGSOC and provides a new

concept toward a yet unmet clinical need. We are planning to further improve our model through further validation and various explainable methods.

3) line 78: "lacks histological grading" -> what does this mean exactly? HGSOC is high grade by definition

As you pointed out, ovarian serous carcinoma is histologically divided into HGSOC and low-grade serous ovarian carcinoma (LGSOC), and HGSOC is already a histologically graded group. However, HGSOC accounts for approximately 75% of epithelial ovarian cancers, and it is well-known for the diverse histologic features⁵. In addition, there are clinically unmet needs to further subclassify HGSOC. Although we intended to emphasize the unmet need for subclassification in HGSOC, the "lack of histological grading," as you pointed out, could confuse the readers. Therefore, we removed the mention about histological grading and revised the sentence as follows.

Before revision: However, unlike other cancers, HGSOC lacks histological grading or independent pathological factors that provide predictive or prognostic information.

After revision (page5, line77): HGSOC exhibits various histopathologic features, but so far, no one has identified pathologic factors that predict clinical outcomes.

4) Please define which dataset is "internal" and which "external" (assuming TCGA is "external" but it would be good to define it explicitly.)

Yes, we addressed your concerns by explicitly defining the internal and external datasets in our revised manuscript, Figures, and Tables. With the inclusion of an additional external cohort

(SMC cohort), special attention was given to articulating concise and reader-friendly descriptions for each dataset, aiming to avoid potential confusion for readers. Here are a few examples of the changed parts:

Page 8 line 155: In the external (TCGA) validation cohort, the PathoRiCH-predicted groups exhibited no significant associations for age, BRCA mutation status, and HRD status, except FIGO stage ($p < 0.001$) (Supplementary Table 3).

Page 13 line 263: To identify the most optimal model that overcomes overfitting to the internal cohort, we compared the performances of six trained models in the external (TCGA) cohort.

5) line 299: this should be "proves" instead of "disproves" I believe?

We revised the sentence based on your comment.

Before revision: This disproves that HGSOC classification using histological images alone is a difficult task for not only human pathologist but also artificial intelligence models.

After revision (page15, line314): This demonstrates the challenge of classifying HGSOC based solely on histological images, which arises in part because HGSOC is already histologically classified as high-grade.

6) line 300: I would refer to "statistically significant results" rather than "remarkable performance"

According to your comment, we changed the wording and also revised the sentence to articulate our intentions more clearly as below.

Before revision: However, the remarkable predictive values of our model suggest that

histological images definitely provide important clues regarding the biological behavior of HGSOC.

After revision (page15, line316): Nonetheless, the statistically significant predictive outcomes from our model indicate that the histological images offer valuable insights into the biological behavior of HGSOC.

Reviewer #2 (Remarks to the Author): Expert in ovarian cancer genomics, imaging, and therapy

The authors introduce a novel deep-learning approach for predicting patient responses to platinum-based chemotherapy. Although similar studies exist (highlighted by authors in the paper), the researchers distinguish their work through the use of UNetPlusPlus model, external validation, and insights into the model's decision-making process. While the paper has value in bringing an important improvement into the field, the results are not outstanding in terms of prediction (AUC-ROC best values 0.7). Moreover, the scientific novelty is limited to method development, and the study provides little new insights into the biology of ovarian cancer. The addition of other data levels, e.g. spatial transcriptomics on the high-score patches, or highly multiplexed staining of the tissues to confirm the bulk RNAseq deconvolution analyses would significantly increase the biological and clinical impacts of the study.

We agree with your comments. In response to your concerns about the model's performance, we initiated a comprehensive re-development of our model. This involved augmenting the training cohorts with an additional 156 samples, bringing the total up to 394, and implementing a 5-fold cross-validation methodology. Additionally, we introduced an extra layer of external validation using new datasets from Samsung Medical Center (Seoul, Korea) (n=136) to

confirm the reliability of our model. As a result of the more intensive model training and validation, the performances of the all-tissue area MIL models in the external (TCGA) cohort were increased, but the cancer-segmented area MILs, which has been chosen as the best model, showed similar AUC-ROC value compared to our original MIL models. Instead, the gap between the internal and external validation was decreased in the cancer-segmented area MIL.

Regarding your second aspect, we agree that our model identified few new insights into cancer biology. However, through an explainable analysis, we identified that the model made a decision considering previously known prognostic features, such as tumor infiltrating lymphocytes. These findings support the reliability of the model's decision, which was another aim of our analysis.

In addition, the newly identified features need to be further substantiated. In this regard, we highly appreciate your suggestion. As you rightly pointed out, spatial transcriptomics are a suitable method to interpret the attention map consisting differently scored patch images from each tumor. We have actually undertaken spatial transcriptomics for several representative cases from our internal (SEV) cohort, with cancer samples from primary and metastatic regions. However, this analysis requires additional time and the results should be analyzed across various aspects, such as tumor heterogeneity. Therefore, we would like to organize this result in a subsequent study. Our ultimate goal is to transition PathoRiCH into clinical settings; and as a part of this journey, additional multicenter validation and in-depth interpretation of the models are essential. In the revised manuscript, we highlighted the necessity and prospective plan of the further studies as below.

After revision (page 16 line 318): To ascertain the robustness of our model and introduce it into clinical practice, additional multicenter validations and in-depth interpretations of the models are essential.

After revision (page 16 line 324): To correlate the histological and molecular features of the PathoRiCH-predicted groups and find unrevealed clues for model decisions, we are conducting spatial transcriptomics for a future study.

1) Choice of Analysis Groups (Lines 113-114): The rationale for selecting two analysis groups, favourable (≤ 12 months) and poor prognosis (> 24 months), lacks explicit mention. The study should transparently clarify this choice based on clinical relevance and existing knowledge.

In clinical practice, the platinum-free interval (PFI) for ovarian cancer is generally assessed using 6- or 12-month cut-off. Of these, the 6-month cut-off is the most widely used, categorizing PFI groups into “platinum-resistant” (PFI ≤ 6 months) and “platinum-sensitive” (PFI > 6 months). However, with clinical necessity, PFI is often subdivided into up to four groups: “platinum-resistant” (PFI ≤ 6 months), “partially platinum-sensitive” (PFI 6–12 months), “platinum-sensitive” (PFI 12–24 months), and “very platinum-sensitive” (PFI > 24 months) ⁶⁻⁸. According to your comment, we re-classified patients according to the clinically relevant categorization.

For binary grouping of patients, we tried both 6-month and 12-month cut-off. With the 6-month cut-off (dividing patients into “platinum-resistant” (PFI ≤ 6 months) and “platinum-sensitive” (PFI > 6 months)), very low proportions (4.2–18.5%) of the platinum-resistant group were identified in all three cohorts. Despite our attempts to balance the two groups by adjusting training set sizes through random sampling, the resulting model exhibited a low AUC-ROC, and notably low F1-score and precision values.

On the other hand, the models using the 12-month cut-off showed better performance in patient stratification. Regarding the proportions of each group in patient cohorts (approximately 3:7

for poor and favorable groups), this cut-off is also clinically more relevant for patient stratification and combination analysis with molecular biomarkers than the 6-month cut-off. As a result, we used the 12-month cut-off, and divided patients with PFI \leq 12 months as poor response group and patients with PFI >12 months as favorable group. According to the revised cut-off, we re-analyzed all of our results. The revised criteria are described in the Methods section of our revised manuscript (page 18, line 363).

2) Cancer Segmentation Model Validation (Breast Invasive Ductal Carcinoma) (Lines 120-123): Due to using a model trained on a different cancer type (breast invasive ductal carcinoma) for cancer segmentation, validating its performance on HGSOc samples is crucial. This step is pivotal since the study eventually opts for the cancer-segmented area at 20 \times magnification for MIL.

To validate the performance of our cancer segmentation model, we performed a concordance analysis between the pathologist-annotated area and the model's cancer segmented area from 10% of cases from the internal (SEV) and the external (TCGA) cohorts. The model demonstrated consistently good concordance across both the internal (SEV) and external (TCGA) cohorts, exhibiting mean Dice coefficients of 0.781 and 0.836, respectively, with the overall mean of 0.804. As we received a similar comment from Reviewer #1, we provided a more detailed response in Reviewer#1's major comment 2.

3) Calculation of Predictive Value per Slide (Lines 169-170): The calculation method for predictive value per slide is not explicitly explained. Elaborating on this calculation is important for comprehending its significance, especially as it is used to predict patients with

multiple slides.

Here is an example for better clarification. Consider a scenario where we have Patient A with three slides. The MIL model predicts each slide independently, yielding distinct outcomes for favorable and poor group prediction, sequentially. Then, following the computation of the average probability for both the favorable and poor group predictions across all slides, the next step involves utilizing the optimal threshold value derived from the false positive rate (FPR) and false negative rate (FNR) ratios. If the overall average probability for the poor group falls below this threshold, the final patient-level prediction result is categorized as “favorable.” Conversely, if it surpasses the threshold, the final prediction is defined as “poor.” This method ensures a patient-level prognostic determination that considers the aggregated probabilities across all slides and employs an optimal threshold for decision-making.

Given the prediction probabilities for each slide:

Slide 1: [0.64, 0.36]

Slide 2: [0.58, 0.42]

Slide 3: [0.72, 0.28]

Calculate the average of the favorable prediction probabilities:

$$\frac{0.64 + 0.58 + 0.72}{3} = 0.64$$

Calculate the average of the poor prediction probabilities:

$$\frac{0.36 + 0.42 + 0.28}{3} = 0.35$$

Now, considering the optimal thresholds for favorable (0.63) and poor (0.37) predictions, compare the average probabilities: The average favorable prediction probability (0.64) is above

the favorable threshold (0.623), and the average poor prediction probability (0.35) is below the poor threshold (0.377). Therefore, based on these calculations, the final prediction class for Patient A is deemed “favorable.” If both the predicted probability for favorable group and the predicted probability for poor group are either below or above the thresholds, the patient group is defined as the prediction probability group with the maximum value between the favorable prediction probability and the poor prediction probability.

4) The selection criteria for HRD: The authors should explain why they have used Takaya et al algorithm for HRD classification. It would be important to show whether the use of improved and clinically used algorithms (PMID 26957554, 36581696) or detection of mutational signatures (PMID 30988514) might improve the separation of the HRD groups, and thus the performance of the prediction.

As you mentioned, there are currently various algorithms for homologous recombination deficiency (HRD) in HGSOc. The most widely used algorithm is that of mychoice HRD, which is FDA-approved. The algorithm utilizes three indicators: LOH (Loss of Heterozygosity), TAI (Telomeric Allelic Imbalance), and LST (Large-scale State Transitions) to calculate the scores, and a cut-off of 42 points is applied to determine the HRD positive or negative group (Telli et al.; PMID 26957554) ⁹. On the other hand, the HRD algorithm by Takaya *et al.*, which we initially applied for HRD prediction, used modified cut-off as ≥ 63 for HRD positivity (PMID 32066851) ¹⁰. The authors proposed that this modification allowed for a more accurate prediction of HRD groups and survival rates in HGSOc. Similarly, Perez-Villatoro *et al.* (PMID 36581696) introduced the “ovaHRDscar” method to evaluate the HRD status more accurately in ovarian cancer ¹¹. Gulhan *et al.* (PMID 30988514) also used different method—mutational signatures to predict the HRD group, defining the “Sig3 group” as HRD positive ¹².

The authors mentioned that this method has a strength in that it can accurately identify HRD groups even in samples with lower mutation counts from targeted sequencing.

In the revised manuscript, we developed the model for predicting HRD status, applying three of the aforementioned HRD algorithms (from Telli *et al.*, Takaya *et al.*, and Perez-Villatoro *et al.*). The performance of our models is shown in Table 1 below. Of these, the method by Perez-Villatoro *et al.* classified 70.8% (201/284) of the TCGA cases as “undefined” or “not evaluated,” which might affect the low performance of our model (Table 2 below).

Table 1. Performance of multiple-instance learning models in predicting homologous recombination deficiency (shown as Supplementary Table X in the revised manuscript)

	HRD status (TCGA 8:2 split for training and test)								
	HRD status (Telli et al.)			HRD status (Takaya et al.)			HRD status (Perez et al.)		
	5×	20×	Multiscale	5×	20×	Multiscale	5×	20×	Multiscale
AUC-ROC	0.524	0.484	0.451	0.469	0.556	0.514	0.357	0.171	0.407
Precision	0.9	0.742	0.648	0.714	0.75	0.586	1	1	0.871
Recall	0.148	0.383	0.968	0.189	0.226	0.32	0.036	0.036	0.964
F1 score	0.254	0.505	0.776	0.299	0.348	0.415	0.069	0.069	0.915

AUC-ROC, area under the receiver operating characteristic curve; HRD, homologous recombination deficiency

Unfortunately, we were unable to predict the “Sig3 group” as reported by Gulhan *et al.*, as their reported codes were not well-reproduced in our laboratory.

Table 2. Homologous recombination deficiency status according to different algorithms

in the TCGA cohort (shown as main Table 1 in the revised manuscript)

	TCGA (N=284)
HRD status (Telli et al.)	
Positive	153 (55.4%)
Negative	114 (41.3%)
Unknown	9 (3.3%)
HRD status (Takaya et al.)	
Positive	140 (49.3%)
Negative	139 (48.9%)
Unknown	5 (1.8%)
HRD status (Perez-Villatoro et al.)	
Positive	68 (23.9%)
Negative	15 (5.3%)
Not evaluated	29 (10.2%)
Undefined	172 (60.6%)

HRD, homologous recombination deficiency

Minor Issues:

1) Performance Comparison of PathoRiCH (Lines 281-282): While the sentence highlights PathoRiCH's superior performance, it's important to note that Supplementary Table 5 displays models with higher AUC values. Providing context on the specific metric used to assess its superior performance, or emphasizing the model's extensive training and validation on a larger dataset, could improve this statement.

Among the previous studies on histology-based prediction models for HGSOC, only Yu *et al.* and Laury *et al.* developed models for predicting Platinum-Free Interval (PFI) similar to PathoRiCH. However, they did not conduct external validations, and an AUC-ROC value was

not provided. Of the remaining three studies, Zeng *et al.* and Wang *et al.* assessed the model performance and presented higher AUC-ROC values than our model, of up to 0.933. However, these studies primarily focused on predicting the overall survival and Bevacizumab treatment response, respectively, different from the PFI predictions of our model. Additionally, they used a single external cohort harboring a small number of cases with tissue microarrays (TMA) images, rather than whole-slide images (WSI). The image sizes of TMA (generally 1x1 mm) corresponded to approximately 0.2–1% of the sizes of WSI (up to 25×20 mm). In addition, TMA images consisted of the most essential cancer component among the WSI images, which were carefully extracted by experienced pathologists. Therefore, the use of TMA cohort might affect the validation results. On the other hand, we validated our model with two independent external cohorts using WSIs.

As you suggested, we aimed to emphasize our model through extensive training and validation using a larger dataset in our revised manuscript. We augmented the size of our training and internal validation cohorts from 238 to 394, which is the largest cohort within the landscape of ovarian cancer prediction models. We also conducted 5-fold cross-validation to our model for more stable estimation of model performance. As a result, the performances of the all-tissue area MIL models were increased, while the cancer-segmented area MILs showed similar performance compared to our original model. We suspect that prediction of platinum-treatment response for HGSOC using histological images alone is a difficult task, as HGSOC is already histologically classified as high-grade. Thus, the model performance might have reached a plateau. Despite the suboptimal model performance, it is noteworthy that our histology-based model outperforms the current molecular biomarkers in patient stratification. Furthermore, combining PathoRiCH and molecular biomarkers was the first attempt to provide an even more powerful tool for the risk stratification of patients.

2) "Additional Tissues" (Line 73): The term "additional tissues" requires clarification. It could be helpful to elaborate on whether this refers to a larger sample size or a different concept.

We intended “additional tissues” as the additional tumor DNA/RNA samples which are required for the molecular testing. We clarified the terminology in the revised manuscript as below.

Before revision: In addition, genomic assays for *BRCA* mutations and HRD status are expensive, entail a long turnaround time, and require additional tissues, making them challenging to implement routinely in every patient with HGSOC, especially in low resource settings.

After revision (page 4, line 72): In addition, genomic assays for *BRCA* mutations and HRD status are expensive, entail a long turnaround time, and require tumor DNA/RNA samples for analysis, making them challenging to implement in every patient with HGSOC, especially in low-resource settings.

3) Supplementary Material Mistakes (Line 298): Typos, like the one found (“insitution”), should be rectified in the supplementary material.

Thank you for pointing out the typo. We conducted a comprehensive review of our revised manuscript with English proofreading to correct any remaining typographical errors.

Additional Considerations

1) Mitigating Impact of Random Splits: The consistency of performance metrics between

external and internal validation of the prediction model demonstrates limited overfitting to the training set. However, it is still important to consider the inherent variability in performance metrics due to the random nature of data in model training, as this variability introduced by chance can impact results. To ensure robust model evaluation, techniques such as cross-validation or repeated experimentation could be considered to minimize the influence of random splits and provide a more stable estimation of model performance.

Taking your comment into consideration, we modified our approach to conduct a 5-fold cross-validation. For internal validation, our revised manuscript now presents the final performance as the average and standard deviation of the model's performance across each fold. Additionally, for external validation, we employed the best model among the 5-folds. The performance of 5-fold cross-validation (Table) is also mentioned in our response to Reviewer #1's major comment #1.

2) Comparative Analysis of Initial Data Distribution and Predictions (Lines 170-175): It might be insightful to compare the initial distribution of data in Kaplan-Meier survival analysis plots for the initial favourable and poor prognosis groups and the data distribution depicted by the predicted groups' figures. Such a comparison could aid in assessing the model's performance and identifying areas that require refinement.

We appreciate your thoughtful input. We included the initial distribution of data for both favorable and poor prognosis groups in the Kaplan-Meier survival analysis plots (Supplementary Figure 3 below). Patients with PFI ≤ 12 months were categorized as the poor response group, while those with PFI > 12 months were designated as the favorable response group.

To better visualize the distribution of the pathoRiCH-predicted outcomes in initial data, we included a series of stacked distribution bar graphs categorized into four PFI groups: platinum resistant (PFI ≤ 6 months), partially platinum resistant (6–12 months), platinum sensitive (12–24 months), and very platinum sensitive (>24 months). The internal (SEV) cohort's PathoRiCH prediction results and its distributions are shown in Supplementary Figure 4 below. For external (TCGA and SMC) cohorts, their categorized groups were cross-referenced against variables such as *BRCA* mutation status, HRD status, and PathoRiCH predictions, showcasing the distribution of favorable and poor outcomes within each subgroup. The stack distribution bar graph for external (TCGA) cohort is shown in Figure 3c below, and the external (SMC) cohort is shown in Supplementary Figure 5b below.

Supplementary Figure 3

Supplementary Figure 4

Figure 3c

Supplementary Figure 5b

3) TCGA Cohort and PARP Inhibitor Information (Lines 110-111): The absence of PARP inhibitor administration information in the TCGA cohort could limit the study's findings. External validation results could be influenced by the unavailability of this data, affecting the model's generalizability to real-world scenarios involving PARP inhibitors.

As described in our original manuscript, the clinical data for the TCGA cohort does not provide information on the administration of PARP inhibitors. However, we identified that the TCGA-OV cohort was established prior to 2010 and clinical data was collected up to August 25, 2010¹³.

Until 2010, the standard treatment protocol was platinum-taxane chemotherapy, and there were few other treatment options available. Then, on December 19, 2014, the PARP inhibitor—olaparib was first approved by the FDA as a maintenance agent for ovarian cancer¹⁴. Based on this, the TCGA cohort is suspected to be PARP inhibitor-naïve. In the revised manuscript, we mentioned this as a limitation in the Discussion section as below.

After revision (page16, line320): Second, the TCGA cohort did not contain information on PARP inhibitor administration. However, the clinical data for the TCGA cohort were collected only until 2010, and PARP inhibitors were not FDA-approved and introduced to ovarian cancer treatment until 2014². Thus, the TCGA cohort is expected to be PARP inhibitor-naïve.

Reviewer #3 (Remarks to the Author): Clinical expert in ovarian cancer and pathology

This study addresses risk stratification of HGSOC using a combination of AI based image analysis with BRCA/HRD status to predict response to chemotherapy. The results are very promising in showing the predictive value of this combined approach for patient prognosis. This addresses a yet unmet clinical need and has the potential of being translated to other types of cancer.

The choice of study cohorts and the division between training and validation sets seems reasonable. The authors have also provided sufficient details in the methodology to help in attempts to reproduce the work or inform other researchers in the field for comparison with other methods. However, the numbers of cases studied are still limited to ascertain the robustness of the system and illegibility of being rolled out to routine practice. This needs to be well highlighted in the discussion and prospective plans to address that suggested to help in planning future studies that build on this work.

Thank you for your positive feedback. In response to concerns raised by you and other reviewers regarding the cohort size, we have expanded our internal cohort by an additional 156 samples, resulting in a total of 394, which is the largest sample size within the landscape of developing ovarian cancer prediction models. To enhance robustness and reliability of our model, we also implemented a 5-fold cross-validation and an additional external validation cohort from Samsung Medical Center (Seoul, Korea) (n=136). Despite expanding our study group, PathoRiCH showed an AUC-ROC of near 0.6 for the internal and external validation cohorts, implying that the model barely differentiated the two groups, at least the gap of AUC-ROC values between the internal and external validation was decreased. This outcome might reflect the inherent complexity in classifying high-grade serous ovarian cancer (HGSOC) based solely on histology, given its pre-existing classification as a high-grade disease. As you commented, to ascertain the robustness of our model, additional multicenter validations and in-depth interpretation of the models, for example, matching attention map results with spatial transcriptomics analysis, are essential. In the revised manuscript, we highlighted the limitation of our model and the necessity of the further studies as below.

After revision (page 16 line 318): To ascertain the robustness of our model and introduce it into clinical practice, additional multicenter validations and in-depth interpretations of the models are essential.

After revision (page 16 line 324): To correlate the histological and molecular features of the PathoRiCH-predicted groups and find unrevealed clues for model decisions, we are conducting spatial transcriptomics for a future study.

References:

1. D'Amato, M., Szostak, P. & Torben-Nielsen, B. A Comparison Between Single- and Multi-Scale Approaches for Classification of Histopathology Images. *Frontiers in Public Health* **10**, (2022).
2. Marini, N. *et al.* Multi-Scale Task Multiple Instance Learning for the Classification of Digital Pathology Images with Global Annotations. in *Proceedings of the MICCAI Workshop on Computational Pathology* 170–181 (PMLR, 2021).
3. Li, B., Li, Y. & Eliceiri, K. W. Dual-stream Multiple Instance Learning Network for Whole Slide Image Classification with Self-supervised Contrastive Learning. in (arXiv, 2021). doi:10.48550/arXiv.2011.08939.
4. Ahmad, I., Yousaf, M., Yousaf, S. & Ahmad, M. O. Fake News Detection Using Machine Learning Ensemble Methods. *Complexity* **2020**, e8885861 (2020).
5. Höhn, A. K. *et al.* 2020 WHO Classification of Female Genital Tumors. *Geburtshilfe Frauenheilkd* **81**, 1145–1153 (2021).
6. Colombo, N. & Gore, M. Treatment of recurrent ovarian cancer relapsing 6–12 months post platinum-based chemotherapy. *Critical Reviews in Oncology/Hematology* **64**, 129–138 (2007).
7. Marth, C. *et al.* ENGOT-ov-6/TRINOVA-2: Randomised, double-blind, phase 3 study of pegylated liposomal doxorubicin plus trebananib or placebo in women with recurrent partially platinum-sensitive or resistant ovarian cancer. *European Journal of Cancer* **70**, 111–121 (2017).
8. Mahner, S. *et al.* Carboplatin and pegylated liposomal doxorubicin versus carboplatin and paclitaxel in very platinum-sensitive ovarian cancer patients: Results from a subset analysis of the CALYPSO phase III trial. *European Journal of Cancer* **51**, 352–358 (2015).
9. Telli, M. L. *et al.* Homologous recombination deficiency (HRD) status predicts response

- to standard neoadjuvant chemotherapy in patients with triple-negative or BRCA1/2 mutation-associated breast cancer. *Breast Cancer Res Treat* **168**, 625–630 (2018).
10. Takaya, H., Nakai, H., Takamatsu, S., Mandai, M. & Matsumura, N. Homologous recombination deficiency status-based classification of high-grade serous ovarian carcinoma. *Sci Rep* **10**, 2757 (2020).
 11. Perez-Villatoro, F. *et al.* Optimized detection of homologous recombination deficiency improves the prediction of clinical outcomes in cancer. *npj Precis. Onc.* **6**, 1–13 (2022).
 12. Gulhan, D. C., Lee, J. J.-K., Melloni, G. E. M., Cortés-Ciriano, I. & Park, P. J. Detecting the mutational signature of homologous recombination deficiency in clinical samples. *Nat Genet* **51**, 912–919 (2019).
 13. Bell, D. *et al.* Integrated genomic analyses of ovarian carcinoma. *Nature* **474**, 609–615 (2011).
 14. Kim, G. *et al.* FDA Approval Summary: Olaparib Monotherapy in Patients with Deleterious Germline BRCA-Mutated Advanced Ovarian Cancer Treated with Three or More Lines of Chemotherapy. *Clinical Cancer Research* **21**, 4257–4261 (2015).

REVIEWERS' COMMENTS

Reviewer #1 (Remarks to the Author):

The authors have taken on board a large number of suggestions, including increasing the size of their training and testing datasets, implementing 5-fold cross-validation or redoing the transcriptomics analysis. This is great to see. They have also been very transparent about their re-analysis, accepting that some parts of their study needed re-thinking and removing them from the revised manuscript (e.g. cell analysis).

My main remaining concern is around the performance, which even with the larger dataset size is 0.59±0.07 on cross-validation. That means that realistically one would not expect to see a performance higher than 0.66, but equally that the lower end of the performance could be as low as 0.52. This is almost in flipping-a-coin territory, and as such the impact of the model remains limited.

Reviewer #2 (Remarks to the Author):

Significant improvements were implemented into the manuscript. The model's performance was bolstered by augmenting training cohorts with an additional 156 samples, incorporating a 5-fold cross-validation methodology, and introducing an extra layer of external validation using datasets from Samsung Medical Center. The authors acknowledged the limited new insights into cancer biology but emphasized the model's reliability through an explainable analysis, identifying its consideration of known prognostic features. Patient grouping was refined based on clinically relevant categorization, adopting a 12-month cut-off for platinum-free interval. Additionally, the cancer segmentation model was validated through a concordance analysis with the pathologist's annotations. The study now incorporates multiple algorithms for homologous recombination deficiency classification. Finally, the study discusses the limitations regarding PARP inhibitor information in the TCGA cohort. In addition, the necessity of additional studies, including spatial transcriptomics and multicenter validations, was highlighted to strengthen the model's clinical application.

However, there are some minor comments that should be addressed:

Minor comments:

1. Differential Expression Analysis Stringency

The assertion that no significantly overexpressed genes were found in the true-predicted cases compared with the false-predicted cases, with an absolute log₂ fold change threshold of >2.5, raises concern. Considering a log₂ fold change of 2.5 implies a nearly 6-fold change, this threshold might be too high and should be reasoned or explored further for optimal stringency. In addition, specifying used constraints for analysis in Methods can be beneficial, as based on Supplementary Figure 8 of Volcano plots, constraints are different (log₂FC and -log₁₀pval are 1 and ~2).

2. Addressing High False Poor Group Predictions

Supplementary Table 6 indicates a considerable proportion (61.2%) of false poor-predicted cases. It would be beneficial to address such limitation and highlight possible reasons behind it and potential strategies for improvement in Discussion.

3. Soft Voting Threshold Clarification

It would be helpful to explicitly mention the threshold used for soft voting in the ensemble technique to ensure the reproducibility of results.

4. Lines 428-432: Rephrasing to ensure better delivery of key point

Consider rephrasing the sentences to ensure better readability.

5. Supplementary Figure 6: Typo in name of graph A

Comments and possible points of revisions

The chosen model exhibits a moderate AUC-ROC ($\sim 0.6 \pm 0.07$), as observed across 5-fold cross-validation and highlighted by reviewers. Despite this, its performance on external validation sets, including those from TCGA and SMC, also hover around 0.6. This consistency suggests stable discriminative performance across both internal and external datasets. However, while the AUC-ROC remains consistent, other performance metrics exhibit variability. On the internal dataset, the F1 score is ~ 0.5 , while on external sets, it ranges from ~ 0.46 to ~ 0.7 . These discrepancies indicate limitations in the model's generalization capability, with performance varying across datasets.

Such model's performance could be influenced by the diversity in internal and external datasets. Internal SEV set has the highest number of samples in very platinum-sensitive group ($\sim 45\%$ from all samples), while in external sets it varies. Plus, most of the samples are with unknown status of BRCA mutation ($\sim 46\%$), while in two external sets - mostly wild-type for BRCA are present ($\sim 93\%$ and $\sim 79\%$). Molecular profiles, such as BRCA mutation status, are known to affect TME characteristics. The imbalance in molecular profiles across datasets could contribute to variations in model performance, as the model may be optimized for specific TME characteristics present in the training data. When applied to external validation sets with different TME characteristics, the model may face challenges in generalizing its learned patterns, resulting in performance variations observed across datasets.

Finally, performance values could be also affected by the choice of thresholds for favourable and poor groups (0.623 and 0.377, respectively (lines 425-426)).

Recommendations:

- *The claim of PathoRiCH better performance compared to current molecular biomarkers should be refocused (for instance, lines 47-48), as the strength of this study lies in effective stratification of patients based on the combination of PathoRiCH+BRCA+HRD.*
- *I would recommend to still show average and standard deviation across 5-folds for all sets across measures of performance. It will help to better evaluate statistical power of model depending on runs, as datasets are clearly imbalanced.*
- *The choice of thresholds of favourable and poor groups should be reasoned well. In addition, to verify whether the performance values are affected by thresholds' choice confusion matrixes could be added, where one could see what group is misclassified the most.*

Questions

1. *Have the other reviewers raised technical issues that you feel are important to address, particularly regarding model performance? Do you disagree with any of their technical criticisms?*

Yes, the issue raised by reviewers is crucial, as model is moderately performing in internal dataset, but also in external ones. In addition, the variability in performance

metrics across different datasets could be highlighted. While the AUC-ROC remains consistent, the variability in F1 scores suggests limitations in the model's generalization capability. Plus, there is the influence of data sets' diversity on model performance, particularly in the prevalence of certain target group and molecular profiles across sets. Additionally, the choice of thresholds for favorable and poor groups could be a potential factor affecting performance.

So, few recommendations were proposed in *Comments* to find out why model's performance is moderate. However, some concerns do not have an explicit answer, like handling the heterogeneity of datasets.

2. Do the other reviewers' comments alter your stance on the conceptual advance and/or novelty of the study?

The reviewers highlighted the main limitation of study. However, independent of moderate model's performance, authors were able to achieve high predictive power of therapeutic response with combined model PathoRiCH+BRCA+HRD. Plus, found by visualization analysis of model's predictions, histologic feature of marked tumor infiltrating lymphocytes in the favorable group is a biologically meaningful feature reported to be associated with treatment response.

Thus, authors should refocus their claims about better performance of PathoRiCH compared to current molecular markers, highlighting that the combination of PathoRiCH+BRCA+HRD has improved performance.

3. Are there any inaccuracies in the other reports? (In our other reports?)

There are no inaccuracies in other reports. The reviewer highlighted a crucial limitation of study.

4. Do you feel that the eventual applicability of the model, and the advance that the study represents as a whole, would be seriously affected by these concerns about performance?

Concerns about model performance could potentially affect the eventual applicability of the model and the overall advance represented by the study. However, the model employment in combination with molecular markers can provide a platform for a better stratification of patients.

Manuscript ID: *NCOMMS-23-33128*

Title: “Histopathologic Image-Based Deep Learning Classifier for Predicting Platinum-based Treatment Response in High-Grade Serous Ovarian Cancer”

We sincerely appreciate the insightful feedback from the reviewers. We would like to resubmit the revised version of the manuscript in the light of the referees' advice. Our point-by-point responses to the reviewers' comments are provided below. We also responded to the reviewer #2's additional recommendations. Please find hereafter our answers (in **blue**) to the reviewers' comments (in **black**).

The main improvements of the revised manuscript are as below:

- 1) **Refocusing manuscript to emphasize the stratification of patients using the combination of PathoRiCH+BRCA+HRD:** The structure of the according manuscript paragraphs and the content of Figure 3 were revised.
- 2) **Re-analysis of differentially expressed genes with a more inclusive threshold:** Upon applying the revised cutoff criteria, we identified differentially expressed genes between each group and the results were included in the revised manuscript.
- 3) **Addressing high proportion of false poor-predicted group:** We addressed this finding as a limitation of our model and highlighted potential reasons and strategies for improvement.

The minor changes, not described in the “Response to the Reviewers' Comments,” are as below:

- 1) **Figure 3:** The parts of the original Figure 3, not related to the combination of PathoRiCH+BRCA+HRD, have been transferred to Figure 4a, Supplementary Figure

3, and Supplementary Figure 5a.

- 2) **Table 3:** As the content of original Table 3 overlapped with the revised Figure 4, the corresponding contents have been transferred to Supplementary Table 4.
- 3) **Table 2:** In the original Table 2, three values—precision, recall, and F1-score for the External (SMC) validation—had been inaccurately recorded, so corrections have been applied.

Response to the Reviewers' Comments

Reviewer #1 (Remarks to the Author):

The authors have taken on board a large number of suggestions, including increasing the size of their training and testing datasets, implementing 5-fold cross-validation or redoing the transcriptomics analysis. This is great to see. They have also been very transparent about their re-analysis, accepting that some parts of their study needed re-thinking and removing them from the revised manuscript (e.g. cell analysis).

My main remaining concern is around the performance, which even with the larger dataset size is 0.59 ± 0.07 on cross-validation. That means that realistically one would not expect to see a performance higher than 0.66, but equally that the lower end of the performance could be as low as 0.52. This is almost in flipping-a-coin territory, and as such the impact of the model remains limited.

Thank you for your constructive feedback. In response to your advice, we had undertaken substantial efforts to enhance our model, including expanding the training and testing datasets and implementing 5-fold cross validation. Despite these enhancements, our model demonstrates an average performance of 0.59 on the internal dataset, with the lowest performing model in the 5-fold validation achieving an AUC-ROC of approximately 0.52, indicating room for further improvement. However, our model consistently successfully stratified different platinum-treatment response groups in three independent cohorts ($p < 0.05$), and visual and genetic explainable analysis support the model's reliability. Over the last 18 months, we have applied various AI models to predict treatment response in high-grade serous ovarian cancer (HGSOC), and only multiple instance learning (MIL) models yielded statistically significant results. Moreover, combining PathoRiCH with molecular biomarkers achieved more powerful risk stratification of patients, showing potential of our model as a

clinically applicable marker. This approach marks the first attempt in the English literature to clinically utilize histologic images of HGSOC for patient stratification.

According to your concerns, we clearly described the suboptimal performance of our model as a main limitation of our study in the Discussion (page 16, line 320). We also refocused all results of our manuscript to highlight the clinical relevance of our model, which, despite not perfect, represents a stepping stone towards the practical clinical use of histologic images for patient stratification in HGSOC.

Reviewer #2 (Remarks to the Author):

Significant improvements were implemented into the manuscript. The model's performance was bolstered by augmenting training cohorts with an additional 156 samples, incorporating a 5-fold cross-validation methodology, and introducing an extra layer of external validation using datasets from Samsung Medical Center. The authors acknowledged the limited new insights into cancer biology but emphasized the model's reliability through an explainable analysis, identifying its consideration of known prognostic features. Patient grouping was refined based on clinically relevant categorization, adopting a 12-month cut-off for platinum-free interval. Additionally, the cancer segmentation model was validated through a concordance analysis with the pathologist's annotations. The study now incorporates multiple algorithms for homologous recombination deficiency classification. Finally, the study discusses the limitations regarding PARP inhibitor information in the TCGA cohort. In addition, the necessity of additional studies, including spatial transcriptomics and multicenter validations, was highlighted to strengthen the model's clinical application.

However, there are some minor comments that should be addressed:

Minor comments:

1. Differential Expression Analysis Stringency

The assertion that no significantly overexpressed genes were found in the true-predicted cases compared with the false-predicted cases, with an absolute log₂ fold change threshold of >2.5, raises concern. Considering a log₂ fold change of 2.5 implies a nearly 6-fold change, this threshold might be too high and should be reasoned or explored further for optimal stringency. In addition, specifying used constraints for analysis in Methods can be beneficial, as based on Supplementary Figure 8 of Volcano plots, constraints are different (log₂FC and -log₁₀pval are 1 and ~2).

As you pointed out, an absolute log₂ fold change > 2.5 in differentially expressed gene (DEG) analysis is a stringent criterion. In addition, the Volcano plots were generated using a less stringent threshold, an absolute log₂ fold change > 1 and $p < 0.01$, leading to potential confusion among readers. Consequently, we re-visited all of the DEG analysis using a more inclusive threshold of an absolute log₂ fold change > 1 and $p < 0.01$.

Upon analyzing the comparison of the “true favorable-predicted” (n = 134) and “false favorable-predicted” (n = 25) groups, we discovered 13 up-regulated genes and 25 down-regulated genes in the “true favorable-predicted” group relative to the “false favorable-predicted” group (Table 1 below). Of these, several up-regulated genes, such as *PRSSI6*, *KLKB1*, and *ACOD1*, were associated with immune response and immunometabolism^{1,2}. Specifically, *PRSSI6* has been previously reported as an immune-related biomarker in ovarian cancer¹. This corroborates the results of the gene ontology analysis that indicated a prevalence of enriched immune response–related genes for the “true favorable-predicted” group.

In the analysis comparing the “true poor-predicted” (n = 19) and “false poor-predicted” (n =

30) groups, we identified 17 up-regulated genes and 26 down-regulated genes in the “true poor-predicted” group (Table 2 below), where there were prevalent up-regulated stromal tissue-related genes, such as *MYO16*, *ANKRD2*, *LRRC14B*, and *MYO7B*, in the “true poor-predicted” group^{3,4}.

In comparison of the “true favorable” (n = 165) and “true poor” (n = 44) groups, *PRSS16*, which is associated with immune response, was again found to be up-regulated gene in the “true favorable” group (Table 3 below).

In the revised manuscript, we incorporated the findings from the DEG analysis into the Result section (page 11, line 221), and added the Supplementary Table 7–9 for further detail. We also specified the used constraints for analysis in the Methods section (page 23, line 484) and the legend of Supplementary Figure 7 (as below).

Table 1. Differentially expressed genes comparing the true and false favorable-predicted response groups

Gene Name	logFC	P	Description
PLEKHG7	-1.404172014	0.0044285	Up-regulated in the true favorable-predicted group
A2ML1	-1.392017999	0.007505	Up-regulated in the true favorable-predicted group
PRSS16	-1.30355742	0.0007804	Up-regulated in the true favorable-predicted group
KLKB1	-1.243715278	0.0035181	Up-regulated in the true favorable-predicted group
ACOD1	-1.190242616	0.0026756	Up-regulated in the true favorable-predicted group
CCDC194	-1.125844595	0.0055183	Up-regulated in the true favorable-predicted group
RAB27B	-1.04532635	0.0057408	Up-regulated in the true favorable-predicted group
RPA4	-1.042391436	0.0009443	Up-regulated in the true favorable-predicted group
ATCAY	-1.042335978	0.0057463	Up-regulated in the true favorable-predicted group
PIP5K1B	-1.032216463	0.0072732	Up-regulated in the true favorable-predicted group
TOP3B	-1.024752168	0.0013414	Up-regulated in the true favorable-predicted group
CASP10	-1.015779877	5.27E-05	Up-regulated in the true favorable-predicted group
NUTM1	-1.009593211	0.0059648	Up-regulated in the true favorable-predicted group
H2BC10	1.017215559	0.0060517	Down-regulated in the true favorable-predicted group
SHD	1.038052918	0.0063617	Down-regulated in the true favorable-predicted group
GNG8	1.067621506	0.0027907	Down-regulated in the true favorable-predicted group
CYTL1	1.164616539	0.0042616	Down-regulated in the true favorable-predicted group
RAB25	1.166476913	0.0086138	Down-regulated in the true favorable-predicted group
PLPPR3	1.196727201	0.0073678	Down-regulated in the true favorable-predicted group

SLC17A8	1.244727212	0.0017404	Down-regulated in the true favorable-predicted group
HTR1E	1.352901979	0.0026498	Down-regulated in the true favorable-predicted group
HMGA2	1.35602366	0.0052951	Down-regulated in the true favorable-predicted group
ELAVL3	1.396410908	0.0014966	Down-regulated in the true favorable-predicted group
SRD5A2	1.404999323	0.005692	Down-regulated in the true favorable-predicted group
CBLN2	1.449043488	0.0064817	Down-regulated in the true favorable-predicted group
MFAP2	1.486214401	7.80E-05	Down-regulated in the true favorable-predicted group
SERPINB4	1.539911646	0.0072901	Down-regulated in the true favorable-predicted group
SERPINB3	1.599318508	0.0026581	Down-regulated in the true favorable-predicted group
PTF1A	1.609705227	0.0009696	Down-regulated in the true favorable-predicted group
SULT1E1	1.612166487	0.0049077	Down-regulated in the true favorable-predicted group
CNTNAP5	1.620384551	0.001402	Down-regulated in the true favorable-predicted group
NLRP4	1.773222139	0.0007576	Down-regulated in the true favorable-predicted group
CDC20B	1.797294694	0.0061188	Down-regulated in the true favorable-predicted group
C20orf85	1.887236702	0.0076161	Down-regulated in the true favorable-predicted group
GAB4	1.900125489	0.0005184	Down-regulated in the true favorable-predicted group
SERPINB2	2.010292418	0.0013508	Down-regulated in the true favorable-predicted group
SERPINB7	2.110034589	0.0008003	Down-regulated in the true favorable-predicted group
NPY	2.611798403	2.30E-05	Down-regulated in the true favorable-predicted group

Table 2. Differentially expressed genes comparing the true and false poor-predicted response groups

Gene Name	logFC	P	Description
GFRA1	2.004871703	0.007911626	Up-regulated in the true poor-predicted group
GPR17	1.842862845	0.000108513	Up-regulated in the true poor-predicted group
SIGLEC12	1.751298834	0.008267055	Up-regulated in the true poor-predicted group
MYO16	1.444646258	0.002482725	Up-regulated in the true poor-predicted group
ANKRD2	1.390121857	0.002484162	Up-regulated in the true poor-predicted group
CGB7	1.359580863	0.002319809	Up-regulated in the true poor-predicted group
CYP26B1	1.358121758	0.001099539	Up-regulated in the true poor-predicted group
SLC4A1	1.289975621	0.00613948	Up-regulated in the true poor-predicted group
LRRC14B	1.266952674	0.006678565	Up-regulated in the true poor-predicted group
MYO7B	1.221040619	0.009179596	Up-regulated in the true poor-predicted group
LIPI	1.197121748	0.003548594	Up-regulated in the true poor-predicted group
FPGT-TNNI3K	1.186292932	0.005445536	Up-regulated in the true poor-predicted group
SYT3	1.152178067	0.002024311	Up-regulated in the true poor-predicted group
NTRK1	1.103672111	0.00511629	Up-regulated in the true poor-predicted group
SEMA6A	1.101484675	0.009882645	Up-regulated in the true poor-predicted group
NEURL3	1.092014338	0.009305472	Up-regulated in the true poor-predicted group
NLRP1	1.068882481	0.001245575	Up-regulated in the true poor-predicted group
NTN4	-1.040217903	0.006091554	Down-regulated in the true poor-predicted group
PGAP4	-1.053281436	0.004010727	Down-regulated in the true poor-predicted group

RAB6B	-1.055321546	0.000285633	Down-regulated in the true poor-predicted group
TM4SF1	-1.092411969	0.000469543	Down-regulated in the true poor-predicted group
CYP4X1	-1.203865816	0.002899108	Down-regulated in the true poor-predicted group
RASL11A	-1.205667528	0.003138161	Down-regulated in the true poor-predicted group
LPAR3	-1.237595142	0.005303612	Down-regulated in the true poor-predicted group
PIPOX	-1.264345932	0.004525906	Down-regulated in the true poor-predicted group
CYP4Z1	-1.319387526	0.008733775	Down-regulated in the true poor-predicted group
WNT5B	-1.341886268	0.005661677	Down-regulated in the true poor-predicted group
SNAP91	-1.355610116	0.008669057	Down-regulated in the true poor-predicted group
TMEM163	-1.363160578	0.000114519	Down-regulated in the true poor-predicted group
MKX	-1.405544154	0.004115931	Down-regulated in the true poor-predicted group
MAOB	-1.413638553	0.001363159	Down-regulated in the true poor-predicted group
MRAP2	-1.435228988	0.005240236	Down-regulated in the true poor-predicted group
NDP	-1.651574438	0.003354513	Down-regulated in the true poor-predicted group
LDLRAD1	-1.757617528	0.002085061	Down-regulated in the true poor-predicted group
TCF21	-1.767344107	0.002353635	Down-regulated in the true poor-predicted group
ST6GALNAC1	-1.882473079	0.004593951	Down-regulated in the true poor-predicted group
RIMS2	-2.069175198	0.007874531	Down-regulated in the true poor-predicted group
CLDN10	-2.138747623	0.002138154	Down-regulated in the true poor-predicted group
CYP4B1	-2.174525337	0.001643222	Down-regulated in the true poor-predicted group
PHOX2A	-2.342395094	0.005534816	Down-regulated in the true poor-predicted group
TOX3	-2.622876245	0.00253797	Down-regulated in the true poor-predicted group
PLCXD3	-2.718814261	0.000628294	Down-regulated in the true poor-predicted group
SOX2	-2.90111624	0.001012191	Down-regulated in the true poor-predicted group

Table 3. Differentially expressed genes comparing the ground truth favorable and poor response groups

Gene Name	logFC	P	Description
PAX6	-1.787073476	0.001754079	Up-regulated in the true favorable group
SOX2	-1.71657696	0.002583119	Up-regulated in the true favorable group
CLDN10	-1.429978506	0.001719702	Up-regulated in the true favorable group
ELAPOR1	-1.410306154	0.003258073	Up-regulated in the true favorable group
PRSS16	-1.319425843	2.83E-05	Up-regulated in the true favorable group
SLC6A14	-1.294685089	0.00823027	Up-regulated in the true favorable group
VTCN1	-1.174596555	0.002345928	Up-regulated in the true favorable group
CYP4B1	-1.159468311	0.005350167	Up-regulated in the true favorable group
CACNA1E	-1.086757848	0.008813405	Up-regulated in the true favorable group
AADAC	-1.001237748	0.002292418	Up-regulated in the true favorable group
OPCML	1.021355835	0.005583704	Up-regulated in the true poor group
MFAP2	1.110976803	9.46E-05	Up-regulated in the true poor group
HMGA2	1.114329623	0.003687919	Up-regulated in the true poor group

GFRA1	1.121733701	0.007748986	Up-regulated in the true poor group
ELFN2	1.167076813	0.003058004	Up-regulated in the true poor group
ADGRB3	1.181447468	0.001978831	Up-regulated in the true poor group
TENM3	1.207764075	0.00597663	Up-regulated in the true poor group
NLRP4	1.239711116	0.004954689	Up-regulated in the true poor group
GAB4	1.245388352	0.005851375	Up-regulated in the true poor group
BTBD17	1.288949025	0.006391659	Up-regulated in the true poor group
IGLON5	1.308812048	0.002475167	Up-regulated in the true poor group
SERPINB7	1.407931464	0.005083348	Up-regulated in the true poor group
DCAF12L1	1.524137449	0.002156476	Up-regulated in the true poor group
SOX11	1.742310507	0.00190633	Up-regulated in the true poor group
NPY	2.284078874	6.08E-06	Up-regulated in the true poor group

Supplementary Figure 7. Volcano plots of the PathoRiCH-predicted and actual ground truth response groups.

a. Favorable-predicted group, contrasting between true and false predictions. **b.** Poor-predicted group, showing expression differences between false and true predictions. **c.** Ground truth data, distinguishing true favorable and true poor groups. Horizontal dotted line: cut-off of $p < 0.01$, vertical dotted line: cut-off of absolute \log_2 fold change > 1 .

2. Addressing High False Poor Group Predictions

Supplementary Table 6 indicates a considerable proportion (61.2%) of false poor-predicted cases. It would be beneficial to address such limitation and highlight possible reasons behind it and potential strategies for improvement in Discussion.

Thank you for highlighting the important issue. As you pointed out, the high rate (61.2%) of false poor-predicted cases suggests that our AI model tends to more strictly recognize features associated with favorable group prediction, while it shows a more lenient approach when it comes to poor group identification. This might be resulted from data imbalance in our training dataset, which contains more favorable response group (66.5%) than poor response groups (33.5%). In fact, we considered this issue during the model development and attempted data balancing through the selective use of favorable group cases. However, the model exhibited inferior performance compared to the model trained on the entire dataset. We considered that the decreased number of total training cases might have influenced the model performance. Thus, it would be necessary to obtain more ground truth poor response group cases as a training dataset to balance two groups.

In the revised manuscript, we addressed the issue of high rate of false poor-predicted cases as a limitation of our model, and we outlined a possible solution involving the recalibrating the training dataset in the Discussion section (page 16, line 326).

3. Soft Voting Threshold Clarification

It would be helpful to explicitly mention the threshold used for soft voting in the ensemble technique to ensure the reproducibility of results.

In the soft voting ensemble model, the threshold was set based on the Youden's J statistic (Youden's index), same as the single-scale and multiscale MIL models.

In the revised manuscript, we added the following sentence (page 22, line 462): “The threshold for the soft voting ensemble model was set using the Youden's Index, with 0.533 for favorable group and 0.467 for poor group.”

4. Lines 428-432: Rephrasing to ensure better delivery of key point

Consider rephrasing the sentences to ensure better readability.

To set the optimal threshold for favorable and poor group classification, we applied the Youden's J Statistic (Youden's Index), as a single statistic to find the point that maximizes the difference between the true positive rate and the false positive rate. In the revised manuscript, the threshold settings were clearly explained. Additionally, we provided the receiver operating characteristic curves of the Youden's Index in Supplementary Figure 9.

Previous: To define the final group based on the favorable and poor group prediction probabilities of the MIL model, we considered the trade-off between the false positive rate and false negative rate, which varies for each threshold, and set the intersection of the trade-off as the threshold. (The favorable group threshold was set to 0.623, and the poor group threshold was set to 0.377.)

Revised (page 22, line 435): To define the final group based on the favorable and poor group prediction probabilities of the MIL model, we set an optimal threshold using the Youden's Index, which could find the point that maximizes the difference between the true positive rate and the false positive rate. The threshold for the favorable group was set to 0.623, and for the poor group, it was set to 0.377 (Supplementary Figure 9).

5. Supplementary Figure 6: Typo in name of graph A

We have corrected the typo in the subtitle of Supplementary Figure 6, graph a: "Cacner-segmented area" was corrected to "Cancer-segmented area".

Recommendations:

- The claim of PathoRiCH better performance compared to current molecular biomarkers should be refocused (for instance, lines 47-48), as the strength of this study lies in effective stratification of patients based on the combination of PathoRiCH+BRCA+HRD.

As the reviewer recommended, we refocused our manuscript to emphasize the stratification of patients using the combination of PathoRiCH+BRCA+HRD. In the revised manuscript, we presented the results of PathoRiCH+BRCA+HRD immediately after the Kaplan-Meier results of PathoRiCH, and subtitled this section “PathoRiCH+BRCA+HRD shows the best PFI prediction ability.” The paragraphs highlight the PathoRiCH+BRCA+HRD combination as the final outcome of our model. Then, comparison of PathoRiCH with other clinical and molecular biomarkers was described separately in the following paragraphs, subtitled “PathoRiCH was identified as an independent prognostic factor.” Likewise, the survival graph for the PathoRiCH+BRCA+HRD combination, which was previously presented alongside other graphs for *BRCA* mutation, HRD status, and *BRCA*+HRD, has been highlighted independently in the revised Figure 3 (shown below). In the discussion part, the PathoRiCH+BRCA+HRD combination was discussed prior to the comparison of various biomarkers. In the revised manuscript, the changes are highlighted on page 3, line 46; page 8, line 145; page 9, line 162; page 14, line 284; and page 17, line 344.

Figure 3. Kaplan-Meier survival analyses and distribution of the true platinum-free interval groups of PathoRiCH+BRCA+HRD in the TCGA external validation cohort.

a, Kaplan-Meier survival plots of patients categorized by combined PathoRiCH, *BRCA*, and HRD results. The combined PathoRiCH, *BRCA*, and HRD significantly differentiated response groups ($p < 0.001$). The favorable-*BRCA*/HRD-positive group displayed the most favorable PFI, and the poor-*BRCA*/HRD-positive and poor-*BRCA*/HRD-negative groups showed the worst PFI. **b**, Distribution of the four PFI groups (platinum resistant (PFI \leq 6 months), partially platinum resistant (6–12 months), platinum sensitive (12–24 months), and very platinum sensitive (>24 months)) by combined PathoRiCH, *BRCA*, and HRD. The colored bars indicate the percentage of predictions for each outcome group (blue for favorable and red for poor), with numerical values within each bars showing the case count for each category. The combined PathoRiCH+*BRCA*+HRD showed significantly different distributions for the four PFI groups ($p = 0.001$).

- I would recommend to still show average and standard deviation across 5-folds for all sets across measures of performance. It will help to better evaluate statistical power of model depending on runs, as datasets are clearly imbalanced.

We agree with the reviewer's comment and have retained Table 2, which shows the 5-fold results of our models.

- The choice of thresholds of favourable and poor groups should be reasoned well. In addition, to verify whether the performance values are affected by thresholds' choice confusion matrixes could be added, where one could see what group is misclassified the most.

We apologize for the insufficient explanation regarding the choice of thresholds. To set the optimal threshold for favorable and poor group classification, we applied the Youden's J Statistic (Youden's Index), a single statistic to find the point that maximizes the difference between the true positive rate and the false positive rate.

To verify whether the performance values are affected by thresholds' choice, we conducted the model using 0.5 as the threshold, which is the midpoint of the sigmoid function, commonly used in binary classification tasks. However, the model showed bias towards classifying into the favorable group in the internal dataset (Table 1 below). When we applied the thresholds calculated by the Youden's Index, 0.623 for the favorable group and 0.377 for the poor group (Figure below), the model resulted in better prediction performance with a sensitivity of 0.702 and a specificity of 0.654 (Table 2 below).

In the revised manuscript, we added the explanation of identifying optimal threshold and provided the receiver operating characteristic curves of the Youden Index in the Methods (page

21, line 436) and the Supplementary Figure 9, respectively.

Table 1. Confusion matrix for internal validation using threshold [0.5, 0.5]

		Predicted		Total
		Favorable	Poor	
Ground truth	Favorable	True favorable-predicted 104 (64.6%)	False poor-predicted 0 (NE)	104
	Poor	False favorable-predicted 57 (35.4%)	True poor-predicted 0 (NE)	57
Total		161	0	161

NE, not evaluable

Figure. Receiver operating characteristic curves of the Youden Index for identifying the thresholds for favorable and poor response groups

a, For predicting favorable response group, 0.623 was identified as the optimal threshold. **b**, For predicting poor response group, 0.377 was identified as the optimal threshold.

Table 2. Confusion matrix for internal validation using threshold by the Youden's Index [0.623, 0.377]

		Predicted		Total
		Favorable	Poor	

		Favorable	Poor	Total
Ground truth	Favorable	True favorable-predicted	False poor-predicted	104
		68 (80.0%)	36 (47.4%)	
	Poor	False favorable-predicted	True poor-predicted	57
		17 (20%)	40 (52.6%)	
Total		85	76	161

Reference:

1. Liu, J., Liu, Y., Yang, C., Liu, J. & Hao, J. Comprehensive analysis for the immune related biomarkers of platinum-based chemotherapy in ovarian cancer. *Translational Oncology* **37**, 101762 (2023).
2. Ding, C. *et al.* Prekallikrein inhibits innate immune signaling in the lung and impairs host defense during pneumosepsis in mice. *J Pathol* **250**, 95–106 (2020).
3. Kemp, T. J. *et al.* Identification of *Ankrd2*, a Novel Skeletal Muscle Gene Coding for a Stretch-Responsive Ankyrin-Repeat Protein. *Genomics* **66**, 229–241 (2000).
4. Ehrlich, K. C., Lacey, M. & Ehrlich, M. Epigenetics of Skeletal Muscle-Associated Genes in the ASB, LRRC, TMEM, and OSBPL Gene Families. *Epigenomes* **4**, 1 (2020).